# Estimating the Event-Related Potential from Few EEG Trials

**Anders Vestergaard Nørskov**[*]                                                    *aveno@dtu.dk*
*Department of Applied Mathematics and Computer Science, Technical University of Denmark*

**Kasper Jørgensen**[*]                                                    *kasperjoergensen3@gmail.com*
*Department of Applied Mathematics and Computer Science, Technical University of Denmark*

**Alexander Neergaard Zahid**                                  *alexander.neergaardzahid@wsa.com*
*Department of Applied Mathematics and Computer Science, Technical University of Denmark*
*WS Audiology, Lynge, Denmark*

**Morten Mørup**                                                    *mmor@dtu.dk*
*Department of Applied Mathematics and Computer Science, Technical University of Denmark*

**Reviewed on OpenReview:** *https://openreview.net/forum?id=c6LgqDhpH0*

## Abstract

Event-related potentials (ERP) are measurements of brain activity with wide applications in basic and clinical neuroscience, that are typically estimated using the average of many trials of electroencephalography signals (EEG) to sufficiently reduce noise and signal variability. We introduce EEG2ERP, a novel uncertainty-aware autoencoder approach that maps an arbitrary number of EEG trials to their associated ERP. To account for the ERP uncertainty we use bootstrapped training targets and introduce a separate variance decoder to model the uncertainty of the estimated ERP. We evaluate our approach in the challenging zero-shot scenario of generalizing to new subjects considering three different publicly available data sources; i) the comprehensive ERP CORE dataset that includes over 50,000 EEG trials across six ERP paradigms from 40 subjects, ii) the large P300 Speller BCI dataset, and iii) a neuroimaging dataset on face perception consisting of both EEG and magnetoencephalography (MEG) data. We consistently find that our method in the few trial regime provides substantially better ERP estimates than commonly used conventional and robust averaging procedures. EEG2ERP is the first deep learning approach to map EEG signals to their associated ERP, moving toward reducing the number of trials necessary for ERP research. Code is available at `https://github.com/andersxa/EEG2ERP`.

## 1 Introduction

Electroencephalography signals (EEG) represent a summation of the electrical activity of neural processes in the brain. Isolating these processes in EEG data is challenging due to poor signal-to-noise and therefore a common approach is to average many repeated stimulus-locked trials in order to extract the associated event-related potential (ERP) (Luck, 2014). Modern techniques often employ complex averaging and filtering methods to improve the extraction and clarity of ERPs (Leonowicz et al., 2005; Bailey et al., 2023a;b). ERPs have been invaluable for investigating a wide range of topics in neuroscience (Luck, 2022) and psychology (Hajcak et al., 2019), and are relevant for brain-computer interface technologies, such as spelling systems for disabled patients (Farwell & Donchin, 1988), and control of humanoid robots, arms, and wheelchairs (Abiri et al., 2019).

In recent years, deep learning has been widely applied in EEG research to discern signals from noise across classification tasks such as emotion recognition (Jafari et al., 2023), sleep staging (Phan & Mikkelsen, 2022),

---

[*]Equal contribution.

detection of epileptic seizures (Nafea & Ismail, 2022), and classification of ERPs (Craik et al., 2019; Li et al., 2020). Representation learning for EEG has also recently been substantially advanced by exploring transformer architectures and contrastive learning procedures (Kostas et al., 2021; Nørskov et al., 2023). Our starting point will be the recently proposed transformer and contrastive learning based CSLP-AE framework (Nørskov et al., 2023), an autoencoder model which uses a specialized reconstruction loss to encode EEG signals into two latent spaces split respectively to optimally characterize subject and task variability. This model has demonstrated strong performance in learning features useful for classifying both subjects and tasks and has shown promising generalizability and zero-shot capabilities.

In this work, we turn the CSLP-AE architecture into a novel EEG denoising framework. Given a subject's few EEG trials, the aim is to estimate the corresponding subject-specific ERP with the ultimate aim of reducing the number of trials necessary to produce reliable ERP estimates. We call this EEG2ERP. This approach aims to make the process more efficient and less burdensome for both researchers and participants requiring fewer repeated trials. A key feature of the proposed framework is the additional uncertainty awareness. In addition to predicting the ERP for a given EEG signal, the model provides variance estimates learned under a Gaussian likelihood while training on bootstrapped ERP targets, which naturally capture sampling variability.

Our objective is to recover interpretable, component-level event-related potentials that scientists routinely analyze to test hypotheses about cognition, perception, and disease. These exploratory studies depend critically on the quality of the ERP waveform, since its amplitude and morphology must be cleanly measurable to serve as reliable dependent variables. For example, Kutas & Federmeier (2011) emphasize the utility of the N400 amplitude in indexing semantic memory retrieval and integration, Brouwer & Hoeks (2013) highlight how N400 and P600 amplitudes map onto distinct cognitive processes within a language comprehension network, and Fields (2023) review the late positive potential predominantly in the context of its amplitude modulations linked to emotional processing and memory. Reliable access to these signal-level measures with fewer trials therefore directly benefits exploratory and analytic neuroscience.

Our research direction is aligned with best practices outlined for ERP research in clinical populations. As noted by Kappenman & Luck (2016), ERP studies in patient groups face the dual challenge of limited trial numbers and the need for highly reliable measures. ERPs are only useful as biomarkers or dependent measures when their amplitudes and morphologies can be cleanly quantified, yet data constraints often compromise this. By improving ERP quality under reduced trial counts, our approach addresses these concerns, thereby supporting exploratory neuroscience and clinical applications in line with established methodological recommendations.

Furthermore, the well-documented dependence of ERP reliability on the number of trials is an unsolved problem. Huffmeijer et al. (2014) demonstrated adequate to excellent test–retest reliability for multiple ERP components only when sufficient trial numbers were available, recommending at least 30 trials for early components such as the VPP and more than 60 trials for later, broadly distributed components such as the P3. Similarly, developmental and lifespan studies have shown that error-related components such as the ERN and error positivity can be reliably measured with as few as six to eight trials, but only under favorable conditions and with carefully controlled tasks (Pontifex et al., 2010; Meyer et al., 2014). Importantly, child ERP guidelines explicitly recommend minimizing trial numbers and building paradigms around the stability constraints of ERP components (Brooker et al., 2019). These converging findings illustrate that ERP reliability is strongly trial-limited and that many populations, including children, clinical groups, and older adults, cannot realistically provide the large number of artifact-free trials typically required.

The proposed framework is evaluated on the ERP CORE dataset (Kappenman et al., 2021) under a challenging zero-shot generalization setting, assessing the model's ability to estimate ERPs from a few EEG trials of unseen subjects. We further apply the EEG2ERP approach to a dataset on face perception (Wakeman & Henson, 2015) considering both the modeling of EEG and magnetoencephalography (MEG) trials. Finally, we evaluate it against strong baselines on the P300 Speller Brain-Computer Interfaces (BCI) dataset (Won et al., 2022) to assess its applicability in a practical setting where few repeated trials are essential to reduce the burden of using the system. We hypothesize that *deep learning methodologies that explore the multivariate structure of EEG signals accounting for subject and task variability can enhance ERP signal recovery*

*using substantially fewer trials than conventional averaging procedures.* To systematically investigate the proposed EEG2ERP in its capabilities of quantifying the ERP as a function of trials, we consider unseen subjects not used during model training and split each subject's trials into two equal sets that are uniformly drawn among the available trials: one set is used to estimate the ERP with the models and baselines, while the other set is averaged conventionally. We then compare these two ERPs to quantify prediction quality.

In keeping with standard ERP methodology, we treat the many-trial average from the held-out trials as a high signal-to-noise reference against which to evaluate prediction quality. This reference is not assumed to represent the true underlying ERP; it remains an empirical estimate that is itself subject to jitter and artifacts. It nevertheless provides a strong benchmark because averaging across many repetitions is the accepted practice for approximating the underlying component with sufficient signal-to-noise ratio. Our evaluation therefore measures how well the model's few-trial estimate approaches this many-trial reference.

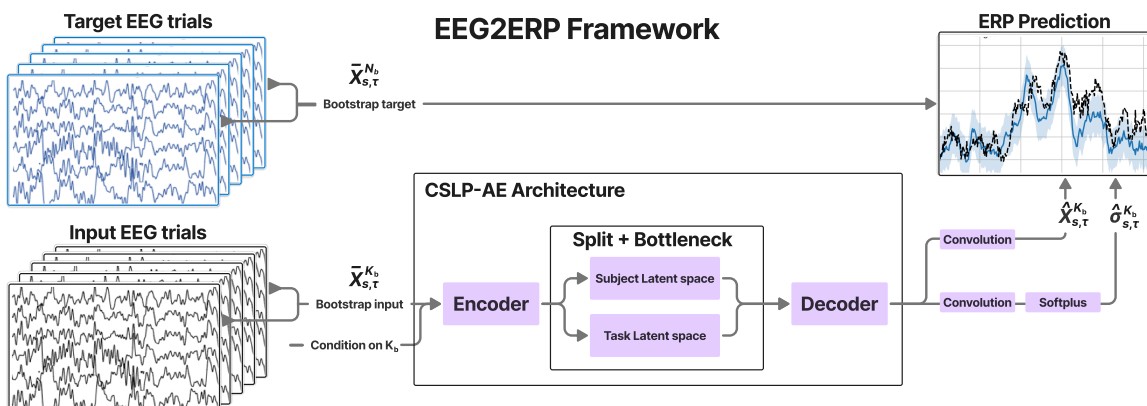

Figure 1: The EEG2ERP model framework. In the top-right graph, the dashed line is the target ERP, and the blue line and area is the model's estimated ERP and confidence respectively. The model maps an input averaged over $K_b$ EEG trials from one set to an estimated ERP averaged over $N_b$ trials from a separate set. It is conditioned on $K_b$ and includes a decoder branch that estimates uncertainty. This visualization simplifies the prediction process to a single channel and demonstrates how bootstrapped ERPs replace the single-trial autoencoder targets of CSLP-AE while accounting for ERP uncertainty including a variance decoder.

## 2 Methods

We denote the set of EEG trials belonging to a specific subject $s$ and task $\tau$ by

$$\mathcal{D}_{s,\tau} = \left\{ \mathbf{X}_{s,\tau}^{(i)} \in \mathbb{R}^{C \times T} \mid i \in \{1, \dots, N\} \right\},$$

where $C$ is the number of channels and $T$ is the number of time points. We use $X_{c,t}$ to denote the value of a trial matrix $\mathbf{X}$ at the $c$'th channel and $t$'th time point.

The averaged ERP over $K$ trials is denoted $\bar{\mathbf{X}}_{s,\tau}^K \in \mathbb{R}^{C \times T}$, and our model's denoised estimate of this ERP is written $\hat{\mathbf{X}}_{s,\tau}^K \in \mathbb{R}^{C \times T}$.

### 2.1 EEG2ERP

We develop an autoencoder framework based on CSLP-AE, that maps an ERP computed from a small number of trials, $\bar{\mathbf{X}}_{s,\tau}^K \in \mathbb{R}^{C \times T}$, to the full ERP obtained from a larger number of trials $\bar{\mathbf{X}}_{s,\tau}^N \in \mathbb{R}^{C \times T}$, where typically $K \ll N$.

The model consists of an encoder, $E_\theta$, and a decoder, $D_\theta$:

$$E_\theta : \mathbb{R}^{C \times T} \to \mathbb{R}^{2 \times C_z \times T_z}, \qquad D_\theta : \mathbb{R}^{2 \times C_z \times T_z} \to \mathbb{R}^{C \times T},$$

where $C$ and $T$ are the number of input channels and time steps, and $C_z$, $T_z$ are the compressed latent dimensions. The bottleneck is split into two latent components: a subject-specific latent $\mathbf{Z}^{\mathbb{S}} \in \mathbb{R}^{C_z \times T_z}$ and a task-specific latent $\mathbf{Z}^{\mathbb{T}} \in \mathbb{R}^{C_z \times T_z}$. These are given as input to the decoder to estimate the ERP. This split-latent structure enables the model to disentangle and retain subject- and task-specific information in the latent space.

Importantly, the encoder acts as an implicit conditioner, as it extracts both subject and task representations at inference time, without requiring explicit conditioning labels. This means that the subject and task information is inferred and embedded within the latent representation during encoding, and is later utilized by the decoder for ERP prediction. As a result, the model can generalize to unseen subjects during testing.

This implicit encoding of subject and task properties makes the framework particularly suited for ERP denoising, where the goal is to preserve meaningful inter-subject and inter-task differences while reducing trial-level noise. We therefore adopt the CSLP-AE approach as the basis for EEG2ERP, with several modifications outlined in the following sections and the appendix.

We reformulate the training objective from conventional autoencoding, as in CSLP-AE, to instead map the average of an arbitrary number of input EEG trials to the associated subject ERP directly. We further augment the training process to account for the uncertainty of the ERP using bootstrapped target ERPs. In addition, we endow the modeling procedure with uncertainty quantification by incorporating a noise variance decoder to quantify the reliability of the estimated ERP. See Figure 1 for an overview of the EEG2ERP procedure.

Specifically, EEG trials are split into two halves: an input half and a target half, sampled uniformly from the available trials before training. Inputs to the model are derived solely from the input half, while target ERPs are constructed from the target half, preventing information leakage between input and target.

Our goal is to evaluate whether the model can predict an ERP that generalizes from trials that were not used to compute the target. Using the full set average as the target while the model sees a subset introduces dependence between predictor and target. The same single trial fluctuations then appear on both sides of the evaluation. This shared noise inflates correlation and reduces error by construction, which produces an optimistic estimate of performance that does not reflect generalization. Split-half evaluation avoids this optimism as the input half and the target half are disjoint, so any agreement can not arise from shared single trial noise.

A further reason for the use of split-halves is parity of signal-to-noise ratio. Comparing one half to the other half equalizes the expected variance on both sides and the power of the average. In addition, when the same trials contribute both to the predictor and to the target average, any latency variability in those trials is shared. This overlap dampens the apparent impact of latency variability, because the predictor and the target are influenced by the same misaligned trials. In contrast, in the split-half setting, latency variability is independent across halves. The agreement in this case cannot be explained by shared jitter, so the evaluation is more sensitive to whether a method truly addresses latency variability rather than benefiting from overlap.

For subject $s$ at task $\tau$, the EEG trials are partitioned into two halves each with $N_{s,\tau}$ trials available. To model variability in ERPs, we use bootstrapping over trials during training (Mooney & Duval, 1993). From the input half, a bootstrap sample of $K$ trials is averaged and encoded into a latent representation. The model then decodes this latent into a predicted ERP and trains against a bootstrap target obtained as a bootstrap sample of $N_{s,\tau}$ target-half trials.

The $b^{\text{th}}$ bootstrap sample of $K$ trials is denoted $\mathcal{D}_b \subseteq \mathcal{D}_{s,\tau}$, and the corresponding ERP is denoted and computed with the point-wise average as follows:

$$\bar{\mathbf{X}}_{s,\tau}^{K_b} = \frac{1}{|\mathcal{D}_b|} \sum_{\mathbf{X}_{s,\tau} \in \mathcal{D}_b} \mathbf{X}_{s,\tau}, \tag{1}$$

where $|\mathcal{D}_b| = K_b$. This bootstrap-based formulation acts as regularization by exposing the model to trial-level variability, helping it learn a distributional mapping instead of overfitting to a fixed ERP.

We sample $K_b$ out of the $N_{s,\tau}$ available trials as follows

$$K_b \sim \text{DiscreteWeighted}\left(w_k \propto \frac{1}{k},\ k = 1, \ldots, N_{s,\tau}\right), \tag{2}$$

which biases training toward using fewer trials on the input side.

This sampling strategy plays a similar role to noise scheduling in denoising autoencoders and diffusion models. By drawing $K_b$ with probability proportional to $1/k$, the model is exposed more frequently to low–trial-count inputs, where averaging provides only weak noise suppression. This emphasizes learning to recover ERPs under high noise conditions, which aligns with our goal of operating reliably when few trials are available. In preliminary experiments, we evaluated several alternative schedules but found that the inverse weighting yielded the strongest performance in the low–trial regime.

In addition to the bootstrap-averaged signal $\bar{\mathbf{X}}_{s,\tau}^{K_b}$, the model is conditioned on the trial count $K_b$ using a sinusoidal positional embedding (Vaswani et al., 2017), which is projected and added before the bottleneck transformer. We could also input the empirical pointwise standard deviation with the average, but this statistic depends heavily on $K_b$, is unstable in the few-trial regime, and is undefined at $K_b{=}1$. Instead, the model learns the per-time-point variance by maximizing the likelihood of bootstrapped target ERPs.

Given an input bootstrap average and trial count, the full encoder–decoder model computes the estimated ERP and uncertainty as follows:

$$\mathbf{Z}^{\mathbb{S}}, \mathbf{Z}^{\mathbb{T}} = E_\theta\left(\bar{\mathbf{X}}_{s,\tau}^{K_b} \mid K_b\right), \tag{3}$$

$$\hat{\mathbf{X}}_{s,\tau}^{K_b},\ \hat{\boldsymbol{\sigma}}_{s,\tau}^{K_b} = D_\theta(\mathbf{Z}^{\mathbb{S}}, \mathbf{Z}^{\mathbb{T}}), \tag{4}$$

where $\mathbf{Z}^{\mathbb{S}}$ and $\mathbf{Z}^{\mathbb{T}}$ are the latent representations for the subject and task latent space respectively, $\hat{\mathbf{X}}_{s,\tau}^{K_b}$ is the estimated ERP, and $\hat{\boldsymbol{\sigma}}_{s,\tau}^{K_b} \in \mathbb{R}^{C \times T}$ is the predicted point-wise standard deviation.

The model is trained to maximize the following likelihood of the bootstrap target ERP from the output half:

$$p_\phi(\bar{X}_{s,\tau,c,t}^{N_b} \mid \mathbf{Z}^{\mathbb{S}}, \mathbf{Z}^{\mathbb{T}}) = \mathcal{N}(\bar{X}_{s,\tau,c,t}^{N_b} \mid \hat{X}_{s,\tau,c,t}^{K_b}, (\hat{\sigma}_{s,\tau,c,t}^{K_b})^2), \quad \forall c, t \tag{5}$$

where $\bar{\mathbf{X}}_{s,\tau}^{N_b}$ is the $b^{\text{th}}$ bootstrap ERP using $N_{s,\tau}$ trials from the target half.

This probabilistic formulation enables the model to learn not only the ERP but also an estimate of its variability over both channels and time for each trial. The noise scale is predicted using a dedicated convolutional branch in the decoder, transformed via the softplus activation to ensure positivity, following standard practice (Kingma & Welling, 2014; Tomczak, 2021).

## 2.2 Loss function

Following Nørskov et al. (2023), the total loss $\mathcal{L}$ is given by combining the reconstruction loss $\mathcal{L}_{\text{R}}$, the contrastive learning loss $\mathcal{L}_{\text{CL}}$, and the latent permutation loss $\mathcal{L}_{\text{LP}}$:

$$\mathcal{L}_{\text{TOT}} = \mathcal{L}_{\text{R}} + \mathcal{L}_{\text{CL}} + \mathcal{L}_{\text{LP}}. \tag{6}$$

These loss terms are substantially modified for the EEG2ERP framework as outlined below.

**Reconstruction loss** $\mathcal{L}_{\text{R}}$  To enable learning of the noise variance, the CSLP-AE $L_2$ reconstruction loss is replaced with a negative log-likelihood loss as used in deep latent variable models. We use a multivariate Gaussian distribution parameterized by the estimated mean $\hat{\mathbf{X}}$ and standard deviation $\hat{\boldsymbol{\sigma}}$:

$$\mathcal{L}_{\text{R}} = \sum_{c=1}^{C} \sum_{t=1}^{T} \left[\frac{\|\bar{X}_{c,t} - \hat{X}_{c,t}\|_2^2}{2\,\hat{\sigma}_{c,t}^2} + \log \hat{\sigma}_{c,t}\right], \tag{7}$$

where $\bar{\mathbf{X}} \in \mathbb{R}^{C \times T}$ is the target ERP, $\hat{\mathbf{X}}$ is the predicted ERP, and $\hat{\boldsymbol{\sigma}}$ is the predicted point-wise standard deviation. This loss enables learning both an estimate of the ERP and its associated uncertainty.

In practice, we stabilize training by linearly annealing the predicted scale from one to $\hat{\boldsymbol{\sigma}}$ over the first half of optimization. This variance annealing strategy is a heuristic generalization of the two-staged optimization approach proposed by Stirn et al. (2023). In that framework, the model is first trained with squared error loss on the predicted mean, which is equivalent up to a multiplicative factor to fixing the variance at one, and only afterwards is the variance estimator optimized using the negative log-likelihood. Our interpolation scheme replaces this discrete two-stage procedure with a smooth transition, linearly annealing the variance contribution from a fixed value of one toward the model's predicted variance over the course of training. This can be seen as a heuristic extension of the faithful training principle, providing a more gradual training signal that stabilizes training.

**Contrastive loss $\mathcal{L}_{\mathrm{CL}}$**  This loss incorporates contrastive learning into each split-latent space (subject or task). Pairs of subjects and tasks are sampled separately, and a contrastive loss is applied to each respective latent space to encourage specialization.

Let $\mathrm{sim}(\mathbf{u}, \mathbf{v}) = \frac{\mathbf{u}^\top \mathbf{v}}{\|\mathbf{u}\|\|\mathbf{v}\|}$ denote cosine similarity. Given $M$ latent pairs $\mathbf{Z}_m^A, \mathbf{Z}_m^B \in \mathbb{R}^H$ in latent space $\mathbb{L} \in \{\mathbb{S}, \mathbb{T}\}$, the full symmetric normalized temperature-scaled cross entropy loss (Sohn, 2016; Oord et al., 2018; Radford et al., 2021) is:

$$\mathcal{L}_{\mathrm{CL}}^{(\mathbb{L})} = \frac{1}{M} \sum_{m=1}^{M} \left[ -\log \frac{\exp\left(\mathrm{sim}(\mathbf{Z}_m^A, \mathbf{Z}_m^B)/\tau\right)}{\sum_{i \neq m} \exp\left(\mathrm{sim}(\mathbf{Z}_m^A, \mathbf{Z}_i^B)/\tau\right)} - \log \frac{\exp\left(\mathrm{sim}(\mathbf{Z}_m^B, \mathbf{Z}_m^A)/\tau\right)}{\sum_{i \neq m} \exp\left(\mathrm{sim}(\mathbf{Z}_m^B, \mathbf{Z}_i^A)/\tau\right)} \right], \tag{8}$$

where $\tau$ is the temperature. To support efficient training, we construct a batch containing all combinations of subject-task pairs. Let $N_S$ and $N_T$ denote the number of subjects and tasks, respectively. All $N_S \times N_T$ combinations are encoded into tensors:

$$\mathbf{Z}^{A,\mathbb{L}}, \ \mathbf{Z}^{B,\mathbb{L}} \in \mathbb{R}^{N_S \times N_T \times H},$$

where $H$ is the latent size, and $\mathbb{L}$ denotes the latent space (subject $\mathbb{S}$ or task $\mathbb{T}$). In cases where the number of subjects or tasks makes such a tensor infeasible (perhaps due to memory constraints), subsampling $N_S$ subjects or $N_T$ tasks works explicitly in this setup.

Similarities are computed by aggregating across the non-corresponding axis to form the following similarity matrices:

$$\mathbf{S}_{ij}^{\mathbb{S}} = \sum_{\tau=1}^{N_T} \mathrm{sim}(\mathbf{Z}_{i,\tau}^{A,\mathbb{S}}, \mathbf{Z}_{j,\tau}^{B,\mathbb{S}}) \quad \forall i,j \in \{1, \ldots, N_S\}, \tag{9}$$

$$\mathbf{S}_{ij}^{\mathbb{T}} = \sum_{s=1}^{N_S} \mathrm{sim}(\mathbf{Z}_{s,i}^{A,\mathbb{T}}, \mathbf{Z}_{s,j}^{B,\mathbb{T}}) \quad \forall i,j \in \{1, \ldots, N_T\}. \tag{10}$$

The resulting subject similarity matrix aggregates similarities across tasks, and the task similarity matrix aggregates similarities across subjects. These matrices are then used within the contrastive loss by using cross-entropy on each similarity matrix and its transpose. This encourages subject representations that correspond to the same individual to cluster in subject-latent space, and likewise encourages task representations that correspond to the same task to cluster in task-latent space, while pushing apart mismatched pairs.

**Latent permutation loss $\mathcal{L}_{\mathrm{LP}}$**  The latent permutation loss is adapted to support general permutations over the latent dimensions instead of fixed pairwise swaps. This generalizes the loss used in standard autoencoders, CSLP-AE, and its quadruplet permutation variant (Nørskov et al., 2023).

Let again $\mathbf{Z}^{A,\mathbb{L}}, \mathbf{Z}^{B,\mathbb{L}} \in \mathbb{R}^{N_S \times N_T \times H}$ denote two realizations of the encoding for all subject-task combinations, as similarly used in the contrastive loss. Permutations are applied along the axis not associated with the current latent space:

- For $\mathbb{L} = \mathbb{S}$ (subject space), apply $\varrho_2(\cdot)$, a random permutation over axis 2 (task dimension).

- For $\mathbb{L} = \mathbb{T}$ (task space), apply $\varrho_1(\cdot)$, a random permutation over axis 1 (subject dimension).

Define $\varrho_i(\cdot)$ as the random-permutation operator on axis $i$. Then the decoder $D_\theta$ takes the permuted latents and outputs an ERP prediction. The permutation loss is then computed as an instance-wise reconstruction loss against the original (non-permuted) ERP:

$$\mathcal{L}_{\mathrm{LP}} = \mathcal{L}_{\mathrm{R}}(\bar{\mathbf{X}}, \ D_\theta(\varrho_2(\mathbf{Z}^{A,\mathbb{S}}), \ \varrho_1(\mathbf{Z}^{A,\mathbb{T}}))) + \mathcal{L}_{\mathrm{R}}(\bar{\mathbf{X}}, \ D_\theta(\varrho_2(\mathbf{Z}^{B,\mathbb{S}}), \ \varrho_1(\mathbf{Z}^{B,\mathbb{T}}))). \tag{11}$$

This loss encourages disentanglement by forcing the model to produce consistent ERP estimates even when the subject and task latents are randomly mismatched ensuring consistent extracted information across the two latent spaces.

**Additional modifications**  Finally, we replace the rectified linear units and convolution strides in CSLP-AE with gated linear units and linear interpolation. These changes are further detailed in Appendix F and G, respectively. Code is available at `https://github.com/andersxa/EEG2ERP` and hyperparameters for the model are detailed in Appendix I.

## 3 Baseline ERP estimation procedure

We evaluate three groups of baseline methods for estimating event-related potentials.

**Sample averaging**  We first compare against conventional averaging across trials. This approach improves the signal-to-noise ratio by reducing random noise as more trials are included, but it assumes that all trials are equally reliable and aligned. When noise varies across trials, the expected decrease in noise proportional to $1/\sqrt{K}$ may not hold. Trials containing outliers, artifacts or other systematic noise can distort the ERP, particularly when only few trials are available. Differences in subject-specific responses or slight timing variability can also produce misalignment, reducing representativeness of the true event-related signal. Robust averaging methods mitigate these effects by down-weighting outliers or aligning trials before averaging. As robust averaging baselines, we use the tanh weighting scheme introduced as a robust location estimator by Leonowicz et al. (2005) and Dynamic Time Warping averaging as described by Molina et al. (2024) (detailed descriptions are provided in Appendix H.1 and Appendix H.2). These methods are model-free and can be applied directly to the set of trials in a bootstrap sample.

**Component aligning**  We include several algorithms that align and decompose trials before averaging. Woody's algorithm (Woody, 1967) performs latency correction by iteratively aligning trials within component-specific latency windows. RIDE (Ouyang et al., 2014) extends this idea by decomposing each bootstrap sample into stimulus-locked, central and response-locked components and aligning the relevant component before averaging. Window definitions and component configurations follow ERP CORE and the RIDE base guide. Full settings are reported in Appendix H.3 and Appendix H.4. On the P300 Speller BCI dataset we additionally apply xDAWN (Rivet et al., 2009), a spatial filtering method operating on two or more components to improve the signal-to-noise ratio. The xDAWN components are obtained from the training data.

**Template-based baselines**  Finally, we include template-based methods. The first baseline uses a task-specific global ERP template formed by the grand average of all training subjects and trials. In addition, we also include a nearest-neighbor baseline where we compute per-subject ERPs across tasks on the training set and perform a task-conditioned nearest-neighbor search in ERP space to select the most similar ERP, which is then used as the prediction.

## 3.1 Performance assessments

To compare the predicted ERPs with the target ERP obtained from the second half of the data (i.e. test trials), we compute the root mean squared error (RMSE) across $B = 200$ bootstrapped input samples for a given subject $s$ and task $\tau$:

$$\text{RMSE}\left(\hat{\mathbf{X}}^{K_{1:B}}, \bar{\mathbf{X}}^N\right) = \sqrt{\frac{1}{B}\sum_{b=1}^{B}\frac{1}{T}\left\|\hat{\mathbf{X}}^{K_b} - \bar{\mathbf{X}}^N\right\|_2^2}, \tag{12}$$

where $\hat{\mathbf{X}}^{K_b}$ denotes the predicted ERP from the $b^{\text{th}}$ input bootstrap of $K$ trials, and $\bar{\mathbf{X}}^N$ is the full target ERP computed from $N = N_{s,\tau}$ trials.

We compare this RMSE curve to that of conventional and robust averaging methods.

To enable comparisons across subjects and tasks with different ERP magnitudes, we also compute the coefficient of determination ($R^2$), which normalizes the reconstruction error:

$$R^2\left(\hat{\mathbf{X}}^{K_{1:B}}, \bar{\mathbf{X}}^N\right) = 1 - \frac{\sum_{b=1}^{B}\left\|\hat{\mathbf{X}}^{K_b} - \bar{\mathbf{X}}^N\right\|_2^2}{\sum_{b=1}^{B}\left\|\bar{\mathbf{X}}^N\right\|_2^2}. \tag{13}$$

This $R^2$ score is bounded above by 1 (perfect estimation), with $R^2 = 0$ corresponding to constant zero predictions. Negative values indicate predictions worse than zero-output baselines.

# 4 Experimental setup

## 4.1 Datasets

We developed and tested EEG2ERP on the ERP CORE dataset (Kappenman et al., 2021), which consists of EEG signals recorded from $C = 30$ channels across 40 subjects, covering 7 different ERP components derived from 6 distinct ERP paradigms: **N170** from the Face Perception Paradigm, **MMN** from the Passive Auditory Oddball Paradigm, **N2pc** from the Simple Visual Search Paradigm, **N400** from the Word Pair Judgment Paradigm, **P3(b)** from the Active Visual Oddball Paradigm, and both **LRP** and **ERN** from the Flankers Paradigm. Each ERP component is measured under two contrasting conditions (e.g., related vs. unrelated), resulting in a total of 14 distinct conditions (or tasks) across the dataset.

We applied minimal preprocessing to the trials, following the ERP cleaning procedures from Script #1 of each component, as outlined in the ERP CORE repository[1]. More information about the paradigms and preprocessing steps can be found in Kappenman et al. (2021).

Additionally, we investigated EEG2ERP on an EEG and MEG dataset originating from face perception experiments (Wakeman & Henson, 2015). These datasets contain, respectively, 70 and 102 channels for the EEG and MEG data sampled at 200 Hz. Each trial belongs to one of three classes: Famous, Unfamiliar, or Scrambled. A detailed description of the preprocessing steps is provided in Appendix I.2.

Finally, we evaluated EEG2ERP on the P300 BCI Speller dataset (Won et al., 2022), a brain-computer interface (BCI) dataset. This dataset contains EEG recordings from 55 subjects performing a visual P300 spelling task based on a $6 \times 6$ character matrix. Each trial involves stimulus flashes (rows and columns), and the resulting ERP is used to detect user intent based on the P300 component. EEG was recorded from 64 channels at a sampling rate of 512Hz. The dataset provides a testbed for assessing ERP estimation in a practical, real-world BCI context, particularly under trial-constrained conditions.

---

[1]See the full ERP CORE procedure here: `https://github.com/lucklab/ERP_CORE/blob/master/ERN/ERN%20Analysis%20Procedures.pdf`

### 4.2 Zero-shot generalization

For the framework to be practically useful, it must support zero-shot generalization: mapping EEG data from previously unseen subjects to their corresponding ERP signals. To this end, the ERP CORE dataset was split such that trials from 28 subjects were used for training, 4 for validation, and the remaining 8 for testing. The Wakeman-Henson EEG and MEG datasets were divided into 11 subjects for training, 2 for validation, and 3 for testing. Finally, the P300 BCI Speller dataset was split into 38 subjects for training, 5 for validation and 12 for testing. The exact subject splits are provided in Appendix I.1.

During ERP comparisons, we focus exclusively on the task-relevant EEG channel identified by Kappenman et al. (2021) for the ERP CORE data, and on EEG channel 065 and MEG channel 098 for the Wakeman-Henson data, as these are reported to exhibit the strongest component-specific responses.

## 5 Results

### 5.1 Evaluation of the extracted ERPs from the ERP CORE data

In Table 1, we compare the performance of ERP estimation using conventional averaging, robust averaging procedures, and the EEG2ERP approach. Our findings indicate that for $K = 1$ and $K = 5$ trials, as well as for $K$ corresponding to 10% of the total trials, the EEG2ERP method provides the best ERP estimates, as quantified by prediction quality measured using $R^2$. Notably, EEG2ERP demonstrates strong performance even with as few as $K = 5$ trials.

Additionally, we observe that EEG2ERP outperforms simple averaging even when using 100% of the trials. However, the dynamic time-warping (DTW) averaging procedure proposed by Molina et al. (2024) achieves the best performance when 100% trials are available.

Table 1: Test set results on the ERP CORE dataset. Standard deviations are computed across 10 initializations of the EEG2ERP model.

| Model / Method | $R^2$ (%) | | | |
| --- | --- | --- | --- | --- |
| | K=1 | K=5 | 10% | 100% |
| EEG2ERP | **5.0 ± 1.6** | **31.7 ± 0.6** | **36.1 ± 0.5** | 47.7 ± 0.4 |
| Simple Averaging | -1696.6 | -382.1 | -193.0 | 34.4 |
| Weighted (Leonowicz et al., 2005) | – | -234.1 | -109.6 | 47.1 |
| DTW (Molina et al., 2024) | – | -124.9 | -56.7 | **50.6** |
| Woody's Algorithm | – | -424.1 ± 153.9 | -236.3 ± 77.7 | -10.9 ± 15.3 |
| RIDE | – | -351.5 ± 106.6 | -207.4 ± 54.6 | -29.8 ± 15.9 |
| Nearest Neighbor | – | 14.5 ± 7.0 | 12.0 ± 7.0 | 42.5 ± 4.4 |
| Global Template | – | – | – | 1.71 |

To account for the high degree of task variability in ERP estimation, the left panel of Figure 2 presents boxplots comparing EEG2ERP (at $K = 5$) and simple averaging at $K = 5$ and at 100% against the target ERP estimated from the other half of the test trials. These boxplots, which illustrate performance across the eight test subjects, reveal substantial variability in ERP recovery quality. In many cases, the ERPs poorly replicate those derived from the other half of the trials (as shown by the green boxplots for simple averaging with 100% of trials). At $K = 5$, EEG2ERP generally yields more reliable ERP estimates, whereas conventional averaging, even with all available trials, sometimes fails as seen, for instance, with the MMN/Deviants and N2pc/Contralateral components.

In the right panel of Figure 2, we assess how well the ERP uncertainty estimates are calibrated to the actual variances of the ERPs. We generally observe that relatively low and high estimated variances correspond to low and high true variances, indicating good calibration.

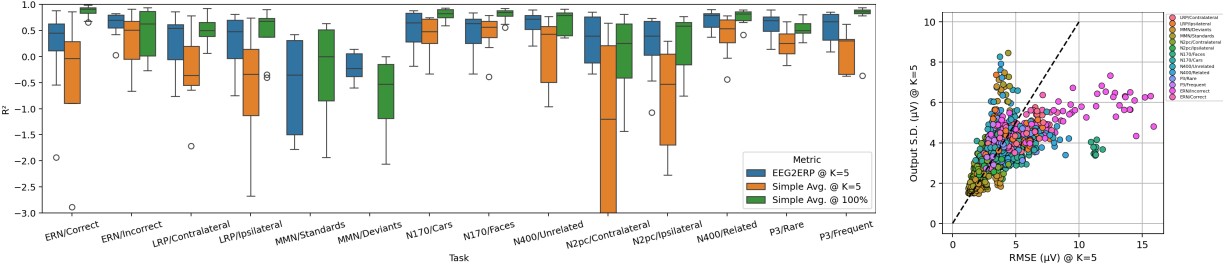

Figure 2: **Left panel:** Boxplots comparing $R^2$ values of the obtained ERPs against the test ERP. Comparisons include the simple average of all trials (100%), simple average over $K = 5$ trials, and EEG2ERP applied at $K = 5$ trials. The missing boxes of simple average at $K = 5$ for the MMN paradigm appear outside the bounds (below $R^2 = -3$). **Right panel:** Variance estimates given by standard deviation as a function of root mean squared error (RMSE) to the ground truth test ERP. Points represent the RMSE of denoised ERPs using $K = 5$ trials to the test ERP for specific subject-task pairs, alongside the noise standard deviation estimates of the denoised trials.

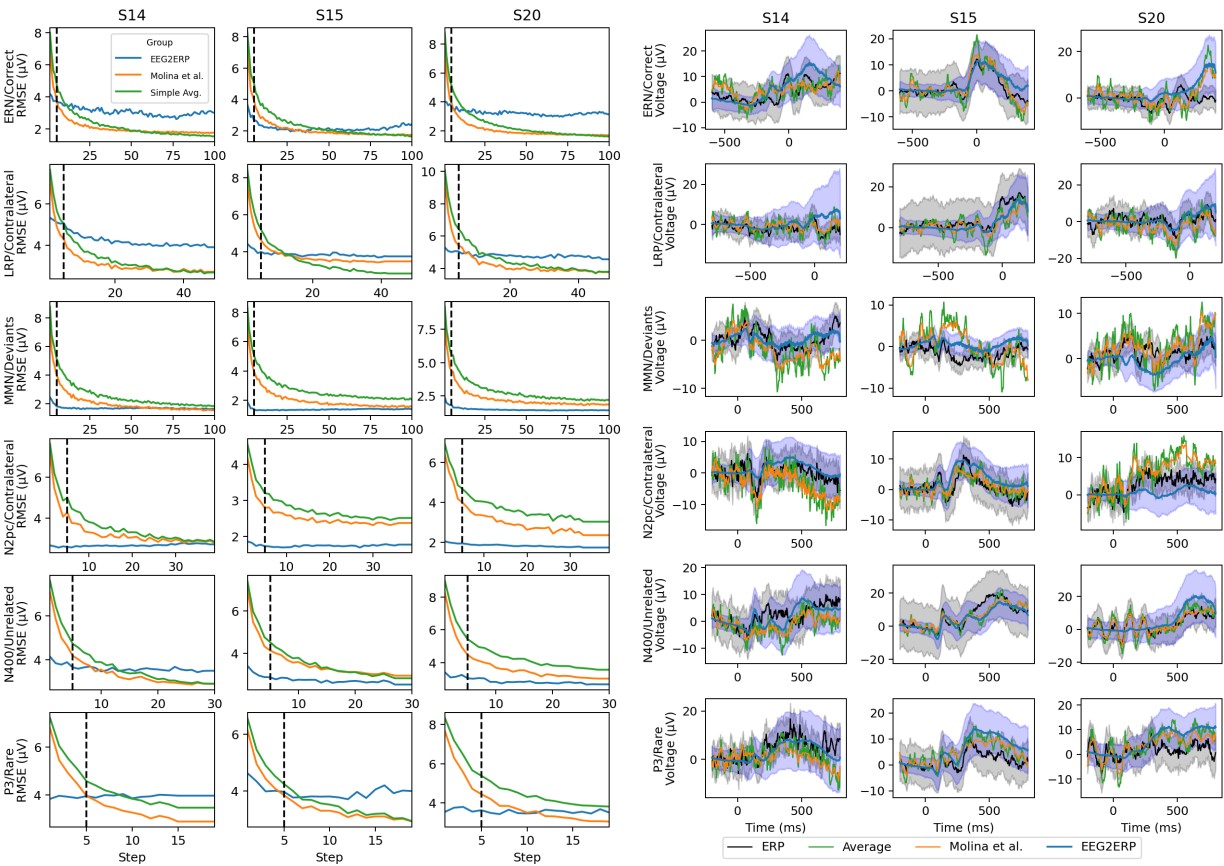

Figure 3: **Left panel:** Root Mean Squared Error (RMSE) as a function of the number of trials included, comparing conventional averaging, Dynamic Time Warped averaging (Molina et al., 2024), and the EEG2ERP procedure. Results are shown for three test subjects across one specific task from each of the six paradigms. **Right panel:** Estimated ERP signals based on $K = 5$ trials, comparing conventional averaging, robust averaging, and EEG2ERP. Confidence bands show bootstrapped estimates of the true ERP, with variance estimates from EEG2ERP predictions displayed as $\pm 2$ times the standard deviation.

The left panel of Figure 3 systematically investigates the ERP estimation quality as a function of the number of trials used. We compare EEG2ERP to both conventional ERP averaging and robust ERP estimation procedures, across three randomly selected test subjects. We find that our denoised estimates (blue curves) outperform both conventional averaging and the best-performing robust method by Molina et al. (2024), particularly in the low-trial regime (vertical black lines mark $K = 5$). Furthermore, the robust estimation methods generally outperform conventional averaging. In the right panel of Figure 3, we also present examples of the corresponding estimated ERPs, including the associated bootstrapped uncertainties and the estimates by the EEG2ERP model.

We examine how well key ERP signal measures reported for the ERP CORE dataset are recovered by EEG2ERP and baseline averaging methods in Appendix A. We also visualize the extracted latent task and subject representations and evaluate their utility for classification in Appendix B. The results indicate that the structure of the latent space improves substantially when using $K = 5$ trials compared to $K = 1$.

To further compare spatial characteristics of the recovered activity, Figure 4 shows topographic maps for a representative (unseen) subject in the N170 Faces paradigm. We visualize scalp potentials at 170 ms and 320 ms after stimulus onset using either one trial, five trials, or all available trials. This allows direct comparison of how EEG2ERP and simple averaging capture physiologically meaningful spatial patterns when little data is available.

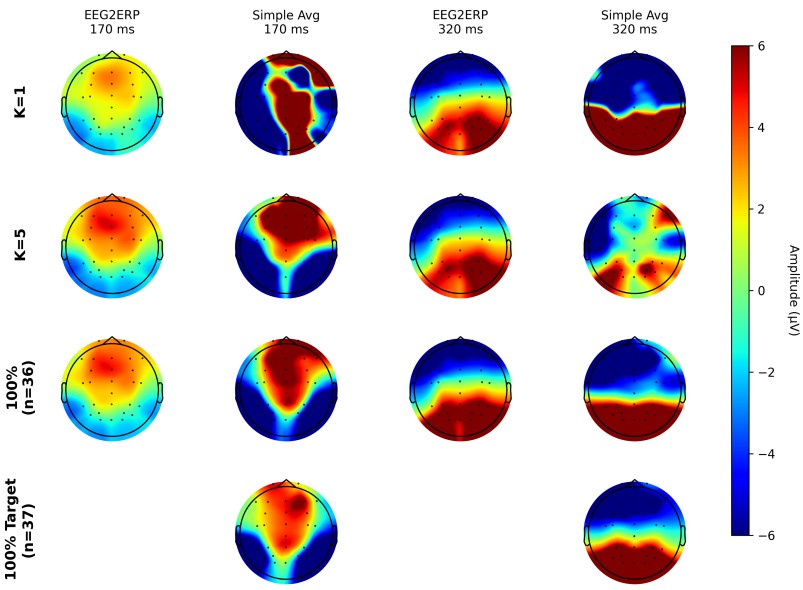

Figure 4: Topographic comparison of EEG2ERP and simple averaging for subject 22, which was not seen during training, under the N170 Faces paradigm at 170 ms and 320 ms. Rows show results using one trial, five trials, all available trials on the model input half, and all available trials on the unseen target half. EEG2ERP recovers the characteristic N170 pattern with very limited data, while simple averaging degrades under low trial counts. Color scale is the same for all plots and indicates scalp potentials in microvolts.

## 5.2 Evaluation of extracted ERPs from the Wakeman-Henson EEG and MEG data

In Table 2, we report the prediction performance for the Wakeman-Henson EEG and MEG datasets. As with the ERP CORE data, EEG2ERP shows favorable performance at $K = 1$, $K = 5$, and when using 10% of the trials, outperforming both conventional and robust averaging methods. However, for the MEG data, the DTW method by Molina et al. (2024) outperforms EEG2ERP when 10% of the trials are used, and all conventional averaging methods outperform EEG2ERP when 100% of the trials are available except the Global Template procedure producing inferior estimates.

Table 2: Test set results on Wakeman-Henson for EEG (top) and MEG (bottom) datasets. Standard deviations are computed across 5 runs of EEG2ERP.

Wakeman Henson EEG Channel EEG065:

| | $R^2$ (%) | | | |
|---|---|---|---|---|
| Model | K=1 | K=5 | 10% | 100% |
| EEG2ERP (Ours) | **16.3 ± 1.6** | **26.7 ± 2.4** | 33.2 ± 2.4 | 38.5 ± 4.0 |
| Simple Averaging | -807.6 | -219.7 | -48.0 | 62.8 |
| Weighted (Leonowicz et al., 2005) | – | -208.7 | -42.3 | 64.4 |
| DTW (Molina et al., 2024) | – | -36.8 | **42.3** | **68.8** |
| Global Template | - - | - - | - - | 19.3 |

Wakeman Henson MEG Channel MEG098:

| | $R^2$ (%) | | | |
|---|---|---|---|---|
| Model | K=1 | K=5 | 10% | 100% |
| EEG2ERP (Ours) | **13.6 ± 2.3** | **19.7 ± 2.2** | **21.8 ± 2.5** | 27.8 ± 5.0 |
| Simple Averaging | -878.6 | -243.0 | -65.8 | 55.5 |
| Weighted (Leonowicz et al., 2005) | – | -246.2 | -67.2 | **56.7** |
| DTW (Molina et al., 2024) | – | -95.1 | 5.7 | 44.2 |
| Global Template | - - | - - | - - | 3.1 |

## 5.3 Evaluation on the P300 BCI Speller Dataset

To further assess the practical utility of EEG2ERP, particularly in the context of Brain-Computer Interfaces (BCIs), we evaluated its performance on the P300 BCI Speller dataset (Won et al., 2022). We compared EEG2ERP against several established ERP estimation and denoising techniques: simple averaging, Dynamic Time Warped (DTW) averaging (Molina et al., 2024), a global ERP template (Template) derived from the training set, and the xDAWN algorithm (Rivet et al., 2009). For xDAWN, spatial components were learned from pooled epochs of all training subjects to enable zero-shot evaluation on the test subjects. The results, measured in terms of $R^2$ (%) against target ERPs constructed from all available trials in a held-out portion of the data, are presented in Table 3.

The results in Table 3 demonstrate that EEG2ERP consistently outperforms other methods in low-trial conditions ($K = 5$ and 10% of trials) for estimating target character ERPs. For instance, at $K = 5$ trials, EEG2ERP achieves an $R^2$ of $33.08 \pm 6.80\%$, substantially outperforming DTW at $-333.25 \pm 112.61\%$ and simple averaging at $-609.79 \pm 181.70\%$. Even compared to xDAWN, a strong spatial filtering baseline, EEG2ERP yields superior performance at low trial counts for target character ERPs. Notably, the Global Template procedure again fails in producing performant ERP estimates.

While xDAWN performs best on non-target ERPs at 100% of trials, EEG2ERP achieves competitive or better $R^2$ scores in few-trial conditions. These results highlight EEG2ERP's robustness in low-data regimes, which are typical in many BCI applications.

## 5.4 Model ablations

In Appendix C, we compare EEG2ERP to a set of ablated variants in order to assess the contribution of key components. These include: (i) CSLP-AE trained on few-trial averaged inputs, (ii) EEG2ERP with single-trial inputs ($K = 1$), (iii) EEG2ERP without uncertainty modeling, and (iv) EEG2ERP without both uncertainty modeling and the trial-count embedding. The single-trial EEG2ERP variant can additionally use ERP variance estimates to weight individual predictions (Appendix E).

Across ablations, CSLP-AE performs poorly when required to estimate ERPs from few-trial averages. The single-trial EEG2ERP variant performs similarly to the full model at $K = 5$ trials but degrades at larger trial

Table 3: Test set $R^2$ (%) results on the P300 BCI Speller dataset (Won et al., 2022). EEG2ERP is compared against Dynamic Time Warped (DTW) averaging, Simple Averaging, a global ERP Template, and xDAWN. $K$ denotes the number of trials used for estimation. Higher $R^2$ values are better.

| Spelling Target Character | | | |
|---|---|---|---|
| Model | $K = 5$ | 10% | 100% |
| EEG2ERP (Ours) | $\mathbf{33.08 \pm 6.80}$ | $\mathbf{34.99 \pm 5.85}$ | $\mathbf{37.28 \pm 5.64}$ |
| DTW (Molina et al., 2024) | $-333.25 \pm 112.61$ | $-65.51 \pm 36.43$ | $21.30 \pm 12.04$ |
| Weighted (Leonowicz et al., 2005) | $-656.42 \pm 193.27$ | $-232.16 \pm 84.65$ | $11.40 \pm 23.55$ |
| Simple Averaging | $-609.79 \pm 181.70$ | $-202.72 \pm 76.71$ | $12.61 \pm 23.06$ |
| Global Template | -- | -- | $32.77 \pm 6.80$ |
| xDAWN (Rivet et al., 2009) | $-129.38 \pm 38.02$ | $-20.10 \pm 17.20$ | $36.46 \pm 7.04$ |

| Spelling Non-Target Character | | | |
|---|---|---|---|
| Model | $K = 5$ | 10% | 100% |
| EEG2ERP (Ours) | $\mathbf{-282.62 \pm 74.91}$ | $\mathbf{-8.86 \pm 15.75}$ | $-13.69 \pm 20.86$ |
| DTW (Molina et al., 2024) | $-3111.61 \pm 903.81$ | $-128.78 \pm 53.74$ | $-6.48 \pm 10.79$ |
| Weighted (Leonowicz et al., 2005) | $-5723.25 \pm 1438.91$ | $-422.03 \pm 123.66$ | $-15.59 \pm 26.00$ |
| Simple Averaging | $-5275.53 \pm 1271.46$ | $-386.58 \pm 111.59$ | $-14.82 \pm 25.78$ |
| Global Template | -- | -- | $-10.23 \pm 16.65$ |
| xDAWN (Rivet et al., 2009) | $-856.37 \pm 206.62$ | $-49.37 \pm 25.90$ | $\mathbf{10.04 \pm 14.19}$ |

counts. Removing uncertainty modeling or both uncertainty modeling and trial-count conditioning leads to reduced performance overall, indicating that both components contribute meaningfully to estimation quality.

Finally, to assess temporal consistency, we conducted an additional experiment where trials were evaluated in chronological order rather than randomly sampled (see Appendix D), confirming that EEG2ERP remains robust under more realistic temporal sampling conditions.

# 6 Discussion

In this work, we introduced EEG2ERP, a deep learning framework for estimating event-related potentials from a small number of EEG trials. This approach enables zero-shot generalization to unseen subjects and incorporates uncertainty estimation by design.

## 6.1 Key Findings

Our experiments consistently showed that EEG2ERP outperforms both conventional and robust averaging methods in the low-trial regime. Remarkably, even with just $K = 5$ trials, EEG2ERP estimated ERPs with higher $R^2$ than conventional averaging using all available trials. This highlights the model's ability to extract meaningful signal representations from noisy input data.

The success of EEG2ERP appears to be driven by two key innovations: (1) introducing a bootstrap-based training strategy that enables robust ERP estimation from varying numbers of trials, with an emphasis on low-trial scenarios, and (2) incorporating uncertainty quantification in the decoder to estimate predictive variability. These components improve robustness and reliability in trial-scarce settings. Combined with the underlying CSLP-AE framework, which enables zero-shot generalization by disentangling subject- and task-specific variability in a split-latent representation, these innovations make it possible to perform low-trial, uncertainty-aware ERP estimation even for subjects unseen during training. Thus, EEG2ERP reduces the number of trials needed to obtain reliable ERPs while generalizing to unseen individuals, offering a powerful and flexible tool for data-efficient neural signal analysis.

## 6.2 Comparison with Prior Work

While previous efforts in ERP estimation have largely focused on signal averaging or spatial filtering (e.g. xDAWN, Template-based Methods), EEG2ERP uses a fundamentally different approach: it learns a encoder-decoder mapping from noisy EEG to clean ERPs in an end-to-end fashion. Unlike robust averaging, which remains model-free and task-agnostic, EEG2ERP benefits from task-aware representation learning and explicit uncertainty modeling. Our results show that this leads to stronger generalization in both EEG and MEG settings, particularly when data is scarce.

Notably, EEG2ERP outperformed established methods like xDAWN and DTW in low-data scenarios but was surpassed by DTW on the MEG data when all trials were used. This suggests that data-rich scenarios may still favor flexible alignment-based methods and traditional denoising, while in contrast, deep models like EEG2ERP remain effective and robust even when trial counts are low and with minimal preprocessing.

We also compared the EEG2ERP procedure to a simple global ERP template. Across the datasets, this template-based ERP estimation consistently produced lower $R^2$ values. This further emphasizes the difficulty of ERP estimation and the advantage of more advanced methods such as the proposed EEG2ERP, particularly in low trial scenarios.

## 6.3 Implications

The ability to estimate ERPs from very few trials opens the door to faster, more efficient EEG-based studies. This could reduce experimental duration, lower participant burden, and enable ERP analysis in populations where many repeated measurements are impractical, such as in children, elderly patients, or individuals with neurological conditions.

Moreover, EEG2ERP provides per-sample uncertainty estimates, which can guide downstream decisions, such as selecting trials for further analysis or identifying unreliable ERPs. This uncertainty-aware feature is especially relevant in clinical and BCI contexts where interpretability and reliability are crucial.

Although EEG2ERP substantially improves amplitude estimation relative to baseline averaging procedures, the gains in latency estimation are more modest. This reflects the fact that our model operates on averaged trial inputs, which partially obscure trial-to-trial latency variability. As a result, latency structure that is misaligned before averaging cannot always be fully recovered. Existing alignment procedures, such as DTW or RIDE, explicitly address latency variability at the trial level, and combining such alignment-based preprocessing with EEG2ERP may produce complementary benefits. Future extensions could also incorporate explicit latency estimation or per-trial encoding, enabling the model to account for variability in timing before aggregation. These directions are promising avenues for improving latency-sensitive ERP characterizations.

## 6.4 Limitations

A key limitation of this work is the variability in model performance across tasks and subjects. Although EEG2ERP performed well on average, specific components (e.g. MMN and N2pc) remain challenging to predict, especially when trial numbers are low or inter-subject variability is high.

Another limitation lies in dataset scope. The datasets presently considered, while comprehensive, represents controlled laboratory settings with well-defined paradigms. Real-world EEG is often more variable and noisy. Future work should test EEG2ERP on more diverse datasets and under naturalistic conditions.

## 6.5 Future Directions

In this work, we built EEG2ERP on top of the CSLP-AE architecture due to its strong zero-shot performance (Nørskov et al., 2023). Future research should explore alternative model architectures, particularly those leveraging self-supervised pre-training or recent EEG foundation models such as BENDR, Neuro-GPT, or LaBraM (Kostas et al., 2021; Cui et al., 2023; Jiang et al., 2024). These approaches could further enhance generalization and robustness, especially in low-data regimes. We presently used a sampling scheme

proportional to the inverse of the number of trials used to emphasize learning in the low trial regime. Exploring alternative sampling schedules that improve performance across the full range of $K_b$ is an interesting direction for future work. Importantly, our work introduces ERP prediction as a novel downstream task for such foundation models emphasizing efficient recovery of evoked responses from very few trials.

## 7   Conclusion

We presented EEG2ERP, a novel deep learning framework for estimating event-related potentials (ERPs) from a small number of EEG trials, with strong generalization to unseen subjects. The method leverages bootstrapped training targets and a learned noise variance decoder to estimate both the ERP and its associated uncertainty. EEG2ERP achieved state-of-the-art performance in few-shot ERP estimation across multiple datasets and modalities, consistently outperforming conventional and robust averaging approaches in low-trial scenarios.

EEG2ERP is the first method to map EEG signals directly to their associated ERPs in a zero-shot setting. Its integration of predictive uncertainty enables more reliable ERP recovery and supports trial selection based on model confidence. As such, EEG2ERP provides not only a powerful new tool for data-efficient ERP estimation but also a valuable benchmark for future EEG representation learning approaches.

We view this work as a foundational step toward more advanced deep learning methodologies in neural signal processing. In particular, the uncertainty-aware design of EEG2ERP is likely to become a key feature in future efforts to reduce the number of trials required for ERP research, enabling broader applicability in both experimental and clinical contexts.

### Broader Impact Statement

This work introduces an approach for estimating ERPs from few EEG trials, which may lower data collection demands and broaden access to high signal-to-noise neural measurements. More efficient ERP extraction could support research in populations that are difficult to study with long repeating protocols and may help enable future applications in adaptive neurotechnology, clinical monitoring, or real-world cognitive assessment.

As with many data-driven models, benefits depend on careful validation and appropriate deployment. Models trained on limited datasets may not capture the full diversity of neural responses, and downstream use should therefore consider the possibility of reduced reliability in underrepresented groups. Continued work on diverse datasets and transparent reporting will help ensure that such tools support inclusive and responsible development across scientific and technological domains.

### Acknowledgments

A. V. N. was supported by Danish Data Science Academy, which is funded by the Novo Nordisk Foundation (NNF21SA0069429) and VILLUM FONDEN (40516). A. N. Z. has received funding from the Lundbeck Foundation (R347-2020-2439).

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

# A ERP CORE metrics

In Table 4 we investigate how well respectively peak latency and amplitude is recovered by the EEG2ERP and traditional averaging procedures, and in Table 5 the recovery of mean amplitude, area latency and onset latency when compared to the reported values of Kappenman et al. (2021).

Table 4: Recovery of peak latency and amplitude
EEG2ERP at K=5 and ERP Core statistics, mean and s.d. over subjects

| Paradigm | Peak Latency (ms) | | Peak Amplitude (µV) | |
|---|---|---|---|---|
| | EEG2ERP | ERP Core | EEG2ERP | ERP Core |
| ERN | 62.18 (12.14) | 54.47 (12.63) | -12.31 (4.91) | -13.86 (7.01) |
| LRP | -64.88 (13.77) | -49.94 (15.66) | -1.62 (1.93) | -3.41 (1.15) |
| MMN | 174.36 (12.01) | 187.60 (19.13) | -1.63 (0.24) | -3.46 (1.71) |
| N170 | 128.99 (9.81) | 131.44 (12.56) | -2.20 (0.72) | -5.52 (3.32) |
| N2pc | 231.13 (13.85) | 253.24 (18.51) | -0.96 (0.81) | -1.86 (1.60) |
| N400 | 372.89 (27.03) | 370.09 (49.43) | -9.29 (1.81) | -11.04 (4.65) |
| P3 | 461.72 (37.06) | 408.89 (70.48) | 6.90 (3.43) | 10.15 (4.53) |

Traditional averaging and Molina et al. averaging for K=5, mean and s.d. over subjects

| Paradigm | Peak Latency (ms) | | Peak Amplitude (µV) | |
|---|---|---|---|---|
| | Simple Avg. | Molina et al. | Simple Avg. | Molina et al. |
| ERN | 55.22 (13.27) | 55.70 (16.16) | -26.15 (9.48) | -20.46 (9.07) |
| LRP | -59.93 (12.37) | -63.34 (11.42) | -8.71 (3.60) | -4.99 (3.75) |
| MMN | 177.94 (7.18) | 176.73 (8.89) | -9.55 (2.67) | -5.05 (1.59) |
| N170 | 132.47 (7.89) | 132.75 (7.82) | -8.33 (2.52) | -5.64 (2.39) |
| N2pc | 234.43 (7.51) | 230.30 (6.13) | -4.79 (1.71) | -2.02 (1.55) |
| N400 | 375.99 (34.16) | 375.85 (36.71) | -17.50 (5.71) | -11.56 (4.83) |
| P3 | 448.23 (30.72) | 441.20 (33.83) | 15.21 (5.21) | 10.89 (4.90) |

Table 5: Recovery of mean aplitude, latency, area latency and onset latency
EEG2ERP at K=5 and ERP Core, mean and s.d. over subjects

| Paradigm | Mean Amplitude (µV) | | 50% Area Latency (ms) | | Onset Latency (ms) | |
|---|---|---|---|---|---|---|
| | EEG2ERP | ERP Core | EEG2ERP | ERP Core | EEG2ERP | ERP Core |
| ERN | -9.31 (3.94) | -9.26 (5.90) | 50.60 (4.42) | 54.36 (9.99) | 10.08 (6.91) | 2.50 (27.53) |
| LRP | 0.58 (2.61) | -2.40 (0.94) | -53.43 (3.35) | -49.09 (9.72) | -88.68 (7.65) | -96.50 (19.85) |
| MMN | -0.58 (0.17) | -1.86 (1.22) | 172.22 (1.67) | 185.20 (14.44) | 148.35 (9.32) | 146.94 (30.33) |
| N170 | -1.09 (0.83) | -3.37 (2.71) | 126.58 (1.74) | 131.84 (8.58) | 117.89 (4.45) | 95.76 (29.05) |
| N2pc | 0.08 (0.87) | -1.14 (1.15) | 237.66 (1.99) | 246.43 (8.81) | 210.21 (4.66) | 213.63 (30.85) |
| N400 | -6.02 (1.77) | -7.61 (3.27) | 390.52 (10.70) | 387.72 (17.85) | 316.02 (13.77) | 284.86 (44.92) |
| P3 | 3.64 (3.48) | 6.29 (3.39) | 453.18 (11.04) | 436.47 (32.77) | 357.00 (25.67) | 327.44 (61.98) |

Traditional averaging and Molina et al. averaging for K=5, mean and s.d. over subjects

| Paradigm | Mean Amplitude (µV) | | 50% Area Latency (ms) | | Onset Latency (ms) | |
|---|---|---|---|---|---|---|
| | Simple Avg. | Molina et al. | Simple Avg. | Molina et al. | Simple Avg. | Molina et al. |
| ERN | -14.60 (9.29) | -12.44 (8.30) | 48.99 (4.18) | 48.37 (4.05) | 16.48 (8.51) | 13.93 (7.99) |
| LRP | 0.13 (4.12) | -0.15 (3.97) | -54.69 (3.44) | -54.00 (3.45) | -82.93 (7.90) | -89.11 (6.73) |
| MMN | -1.23 (2.43) | -1.21 (0.83) | 173.84 (0.75) | 173.99 (1.51) | 149.68 (6.71) | 145.01 (9.17) |
| N170 | -3.72 (1.81) | -2.83 (1.89) | 128.41 (2.08) | 128.75 (2.26) | 121.41 (6.46) | 120.62 (6.40) |
| N2pc | 0.62 (1.73) | 0.70 (1.61) | 236.11 (1.47) | 236.21 (1.89) | 215.56 (5.07) | 210.11 (3.55) |
| N400 | -6.86 (5.20) | -5.46 (4.45) | 395.46 (8.06) | 397.57 (5.81) | 323.48 (11.69) | 320.11 (11.92) |
| P3 | 4.34 (4.02) | 4.32 (3.69) | 456.30 (9.87) | 454.79 (9.40) | 355.33 (20.58) | 353.90 (24.78) |

## B   Model latent space

Figure 5 contains confusion matrices from encoding every trial as a single-trial in the test dataset using the encoder and using the XGBoost Classifier (Chen & Guestrin, 2016) just as in Nørskov et al. (2023). The corresponding subject and latent spaces are shown using $t$-SNE plots in Figure 6.

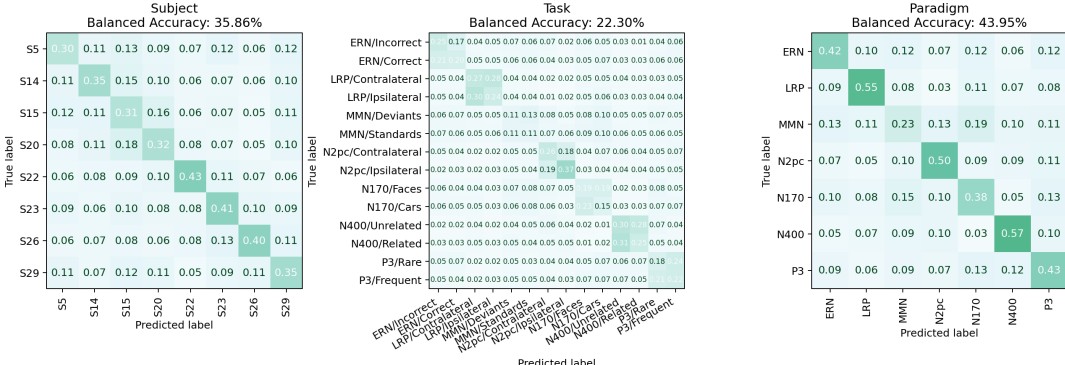

Figure 5: Confusion matrix on subject, task and paradigm labels using XGBoost Classifier with 5-fold Cross-Validation splits as in CSLP-AE. Latents are encoded using single-trial samples only.

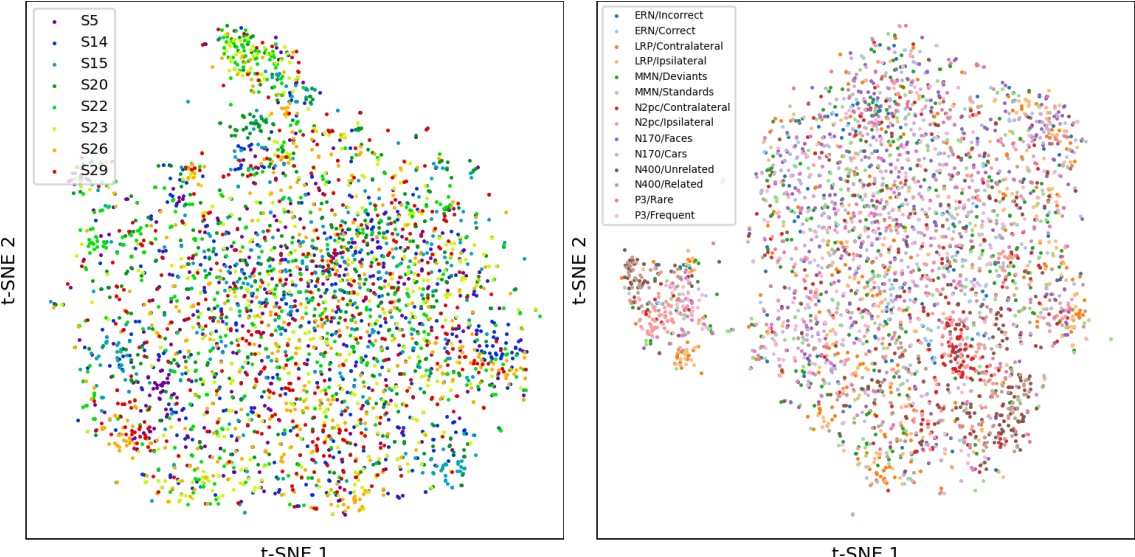

Figure 6: $t$-SNE plot on the subject latents (left) and task latents (right) from single-trial samples.

Figure 7 contains confusion matrices from encoding bootstrap samples of five trials each, without replacement for all available trials to keep samples independent, on the test dataset and using XGBoost to classify. The corresponding subject and latent spaces are shown in Figure 8.

Figure 9 contains confusion matrices from the EEG2ERP model trained with single trial input. Here encoding single trials on the test dataset and using XGBoost to classify. The corresponding subject and latent spaces are shown in Figure 10.

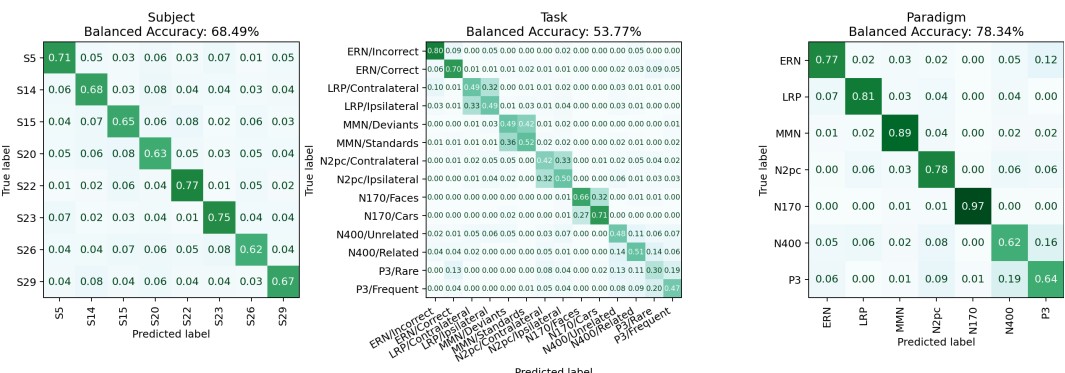

Figure 7: Confusion matrix on subject, task and paradigm labels using XGBoost Classifier with 5-fold Cross-Validation splits as in CSLP-AE. Latents are encoded using five independent trials that are also independent of other samples.

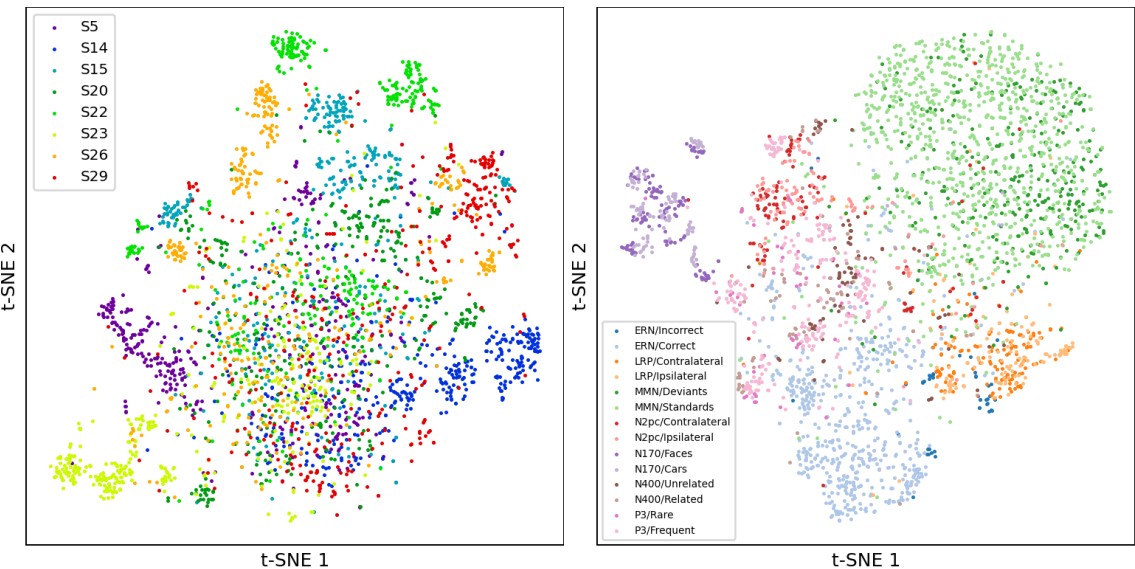

Figure 8: t-SNE plot on the subject latents (left) and task latents (right) from latents with five independent trials that are also independent of other samples.

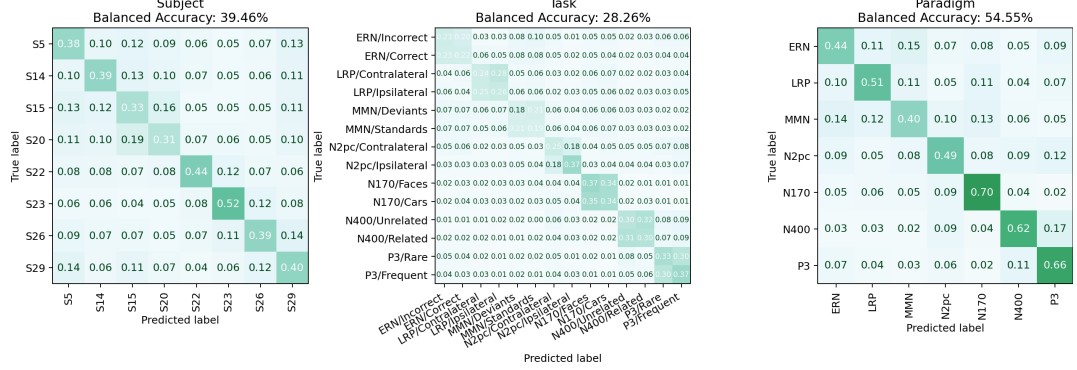

Figure 9: Confusion matrix on subject, task and paradigm labels using XGBoost Classifier with 5-fold Cross-Validation splits as in CSLP-AE. Latents are encoded using single trials using the EEG2ERP trained with single trial input instead of bootstrap.

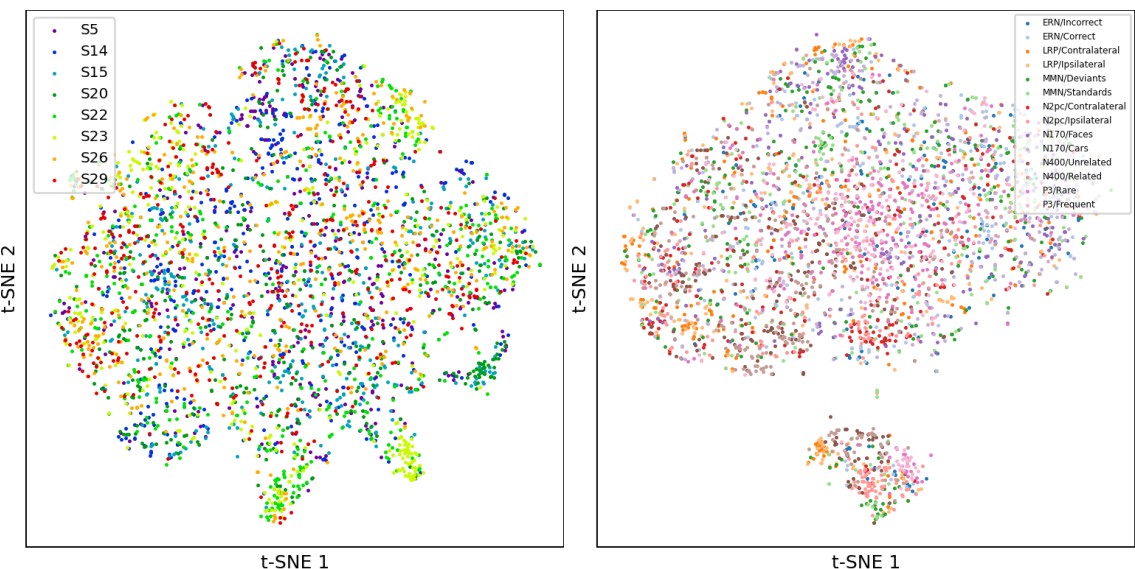

Figure 10: t-SNE plot on the subject latents (left) and task latents (right) from latents with single trials using the EEG2ERP trained with single trial input instead of bootstrap.

## C   Results of CSLP-AE and ablations of EEG2ERP on denoising

We evaluate how different components of EEG2ERP contribute to ERP reconstruction by comparing the full model to several ablated variants. First, we consider a standard CSLP-AE model trained on ERP CORE and evaluated by averaging its single-trial reconstructions on the test set. This baseline does not incorporate uncertainty modeling, trial-count information, or bootstrapped averaging during training. We additionally trained a variant of EEG2ERP using only single-trial inputs ($K^{\text{input}} = 1$). Although this version is not exposed to averaged inputs during training, it can produce an ERP estimate for each trial and combine them using uncertainty-based weighting (Appendix E). This variant is denoted EEG2ERP w/ Single Trial.

In addition to the single-trial ablations, we include two ablations on the uncertainty decoder and the trial-count conditioning. The first removes the uncertainty decoder (EEG2ERP w/o Unc.), which prevents the model from estimating per-time-point variance but retains trial-count conditioning. The second removes both uncertainty modeling and trial-count conditioning (EEG2ERP w/o Unc. w/o Emb.), isolating the effect of the latent-conditioning mechanism on performance. Finally, we evaluate CSLP-AE when the input consists of few-trial averages instead of single trials (CSLP-AE w/ Avg. Input), to assess whether CSLP-AE can learn the mapping without the architectural changes and the uncertainty modelling.

Quantitative results are presented in Table 6. The standard CSLP-AE baseline performs poorly across all conditions, and providing few-trial averaged inputs does not resolve this and only seems to exacerbate it. The single-trial variant of EEG2ERP performs comparably to the full model at $K = 5$ trials but degrades at larger trial counts, reflecting that the full model benefits from being trained on inputs spanning a range of $K$. Removing uncertainty modeling or both uncertainty and trial-count conditioning leads to consistent decreases in performance, particularly when few trials are available. These comparisons indicate that both uncertainty estimation and trial-count conditioning play meaningful roles in robust ERP prediction.

A boxplot showing $R^2$ at $K = 5$ for each of the models above and in Table 6 is shown in Figure 11.

Table 6: Test set results for CSLP-AE with averaging, EEG2ERP with single-trial input, and EEG2ERP.

| Model | | | $R^2$ (%) | |
|---|---|---|---|---|
| | $n$ | $K = 5$ | 10% | 100% |
| CSLP-AE | | -37.3 | -25.0 | -8.6 |
| CSLP-AE w/ Avg. Input | 4 | $-124.4 \pm 0.7$ | $-119.0 \pm 0.8$ | $-99.8 \pm 0.8$ |
| EEG2ERP w/ Single Trial | 6 | $\mathbf{32.6 \pm 0.9}$ | $33.5 \pm 0.7$ | $37.6 \pm 0.7$ |
| EEG2ERP | 10 | $31.7 \pm 0.6$ | $\mathbf{36.1 \pm 0.5}$ | $\mathbf{47.7 \pm 0.4}$ |
| EG2ERP w/o Unc. | 5 | $9.3 \pm 2.2$ | $24.0 \pm 2.9$ | $38.5 \pm 2.5$ |
| EG2ERP w/o Unc. w/o Emb. | 5 | $10.8 \pm 1.9$ | $24.5 \pm 1.9$ | $41.2 \pm 1.5$ |

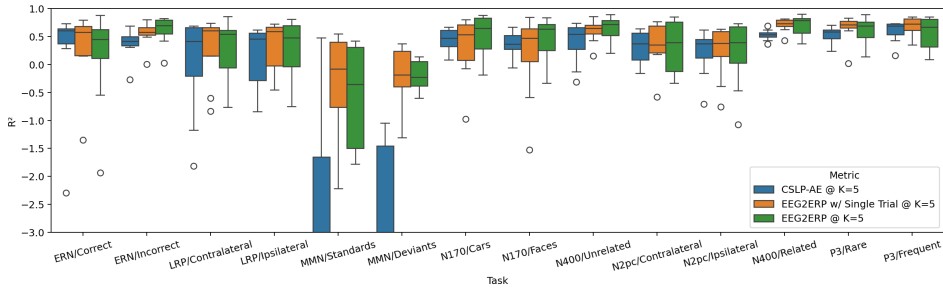

Figure 11: Box-plot of the ablations on the EEG2ERP model, showing the standard CSLP-AE as used for denoising, EEG2ERP trained with single trial input, and the EEG2ERP model.

# D Chronologically sequential trials

We performed an additional experiment evaluating all methods on chronologically sequential trials rather than random samples. In practice, this setting more closely reflects the real-world use case of reducing the number of trials by omitting later ones, thereby preserving their temporal order. Specifically, EEG2ERP was trained as before using randomly ordered bootstrap samples but evaluated using sequentially ordered trials. The results on the ERP CORE dataset are summarized in Table 7.

Table 7: Performance on ERP CORE when using chronologically sequential trials rather than randomly sampled trials. Higher $R^2$ indicates better ERP estimation.

| Model | $n$ | $K = 5$ | 10% | 100% |
|---|---|---|---|---|
| EEG2ERP | 9 | $\mathbf{32.7 \pm 2.7}$ | $\mathbf{43.7 \pm 0.8}$ | $50.4 \pm 0.5$ |
| DTW | | -65.6 | -66.7 | 59.9 |
| Weighted | | -147.9 | -135.0 | $\mathbf{65.3}$ |
| RIDE | | -232.5 | -133.5 | -16.7 |
| Woody's alg. | | -226.3 | -209.1 | 18.6 |
| Nearest neighbor | | 25.7 | 13.3 | 47.7 |
| Simple avg. | | -171.6 | -159.0 | 60.7 |

Overall, results show improved performance across most methods compared to random sampling, which is expected since temporally adjacent trials are less affected by non-stationarities such as changes in attention or fatigue. Importantly, EEG2ERP maintains strong performance under this temporally ordered evaluation, confirming its robustness across different sampling strategies.

# E   Estimation of average ERP from denoised single-trial EEG estimates

To obtain a final ERP prediction from a set of $N$ estimated ERPs obtained from single trial EEG signals by the EEG2ERP, we employ an estimator that maximizes the likelihood of all of the produced single trial ERP estimates (below denoted $\hat{x}_{s,\tau}^{(i)}(c,t)$ for the $i^{th}$ trial for subject $s$ at task $\tau$ for channel $c$ at time $t$ with associated estimated uncertainty $\hat{\sigma}_{s,\tau}^{(i)}(c,t)$). For the ERP estimate of channel $c$ at timepoint $t$ we obtain

$$\hat{\mu}_{s,\tau}(c,t) = \arg \max_{\hat{\mu}_{s,\tau}(c,t)} \prod_{i=1}^{N} \mathcal{N}\left(\hat{x}_{s,\tau}^{(i)}(c,t) \mid \hat{\mu}_{s,\tau}(c,t), \left(\hat{\sigma}_{s,\tau}^{(i)}(c,t)\right)^2\right). \tag{14}$$

The solution to this maximization reduces to calculating a weighted average of the estimated ERPs, where the weights are the inverse variances (precisions) of the signals as given by

$$\hat{\mu}_{s,\tau}(c,t) = \frac{\sum_{i=1}^{N} \left(\hat{\sigma}_{s,\tau}^{(i)}(c,t)\right)^{-2} \hat{x}_{s,\tau}^{(i)}(c,t)}{\sum_{i=1}^{N} \left(\hat{\sigma}_{s,\tau}^{(i)}(c,t)\right)^{-2}} \tag{15}$$

This inverse-variance weighted average can be shown to have the least variance among all weighted averages (Hartung et al., 2008), and the variance of this weighted average estimator is given by:

$$\hat{\sigma}_{s,\tau}^2(c,t) = \mathrm{Var}(\hat{x}_{s,\tau}(c,t)) = \frac{1}{\sum_{i=1}^{N} \left(\hat{\boldsymbol{\sigma}}_{s,\tau}^{(i)}(c,t)\right)^{-2}}. \tag{16}$$

As the variance of the mean is reduced by a factor of $1/N$ we can multiply this variance of the mean by $N$ to obtain an estimate of the average variance of this weighted average ERP estimate. Notably, this averaging procedure thereby accounts for the precision of each individual ERP estimate and gives more weight to single trial estimates with low as opposed to high variance.

# F   Gated Linear Units as opposed to ReLU

The CSLP-AE model architecture utilizes ReLU activations within the convolutional blocks. While ReLU activations are widely used for their simplicity, effectiveness, and fast implementation, recent advancements suggest that Gated Linear Units (GLUs) can provide improved performance in sequence modeling tasks by introducing element-wise gating mechanisms that allow more expressive control over the flow of information (Dauphin et al., 2017; Shazeer, 2020). Consequently, the ReLUs in the ConvBlocks of the CSLP-AE architecture have been replaced with GLUs instead.

The ReLU activation function is defined as:

$$\mathrm{ReLU}(z_{i,j}) = \max(z_{i,j}, 0) \quad \forall i \in \{1, \dots, C_z\}, \forall j \in \{1, \dots, T_z\} \tag{17}$$

where $z_{i,j}$ represents the output of the convolution operation, $C_z$ is size of the latent dimension and $T_z$ is the time-resolution of the latent space.

In contrast, the GLU activation function utilizes a gating mechanism. The convolution output is split into two parts: one-half represents the feature signal, and the other half functions as the modulating gate. For an input $\mathbf{X} \in \mathbb{R}^{C \times T}$ the convolution is expressed as:

$$z_{i,j}^{(f)} = \mathrm{Conv1D}(\mathbf{X})_{i,j}, \quad \text{and} \quad z_{i,j}^{(g)} = \mathrm{Conv1D}(\mathbf{X})_{i,j} \tag{18}$$

However, in practice, this is computed with a single convolution operating with double the filters and with the output channels split evenly. The modulation gate, $\boldsymbol{z}^{(g)}$, is normalized using affine instance normalization (Ulyanov et al., 2016) before applying the activation: $\bar{\boldsymbol{z}}^{(g)} = \mathrm{InstanceNorm1D}(\boldsymbol{z}^{(g)})$. The GLU activation function is then applied as:

$$\mathrm{GLU}(z_{i,j}^{(f)}, \bar{\boldsymbol{z}}_{i,j}^{(g)}) = z_{i,j}^{(f)} \cdot \sigma(\bar{\boldsymbol{z}}_{i,j}^{(g)}) \tag{19}$$

where $\sigma(\cdot)$ is the standard logistic sigmoid function.

In the CSLP-AE architecture, instance normalization is applied directly to the feature signal after each convolution. While this improves training stability by normalizing the input distribution, it also removes scale information from the feature signal. This can adversely affect signal representation. The normalization is applied on an instance basis. This means that normalization is performed at each timestep in the resolution of the latent space. Relative magnitude information between representations in time is thus being removed from the signal by normalizing each such instance.

Applying the normalization function in the gating function is a contribution of this work. By moving instance normalization to the gating mechanism, the derived GLUs maintain scale information in the feature signal through the feature signal path. Here scale information refers to the relative magnitude of features, which can be critical for encoding the strength or importance of specific EEG signal components. For example, the amplitude of an ERP signal reflects physiological characteristics and contributes to its interpretability. Normalization eliminates this magnitude by standardizing feature values for each instance. This loss of scale can hinder the model's ability to leverage amplitude-based patterns in the data.

Instance normalization in the derived GLUs includes learnable affine parameters that can control the behavior of the sigmoid gating function. This enables the model to dynamically modulate the location, sharpness and width of the sigmoid gating function, allowing finer control over information flow and being able to adapt to varying data distributions during training.

With non-linearity confined to the gating operation, the derived GLUs ensure a linear pathway for the residual feature signal between network layers which enhances signal propagation and gradient flow. The feature signal $z^{(f)}$ remains unaltered by non-linear transformations, allowing it to retain structure and scale. This preserves the integrity of the input signal while enabling selective attenuation or amplification in the network by the modulating gate.

## G   Interpolated Residual Connections

The CSLP-AE architecture (Nørskov et al., 2023) is based loosely on the ResNet architecture from He et al. (2016) with residual connections between blocks of convolutions. Residual connections are a core component of deep learning architectures which enables better gradient flow and improved training of deep networks.

In the original ResNet design, when the feature maps of the input and output have different sizes (e.g., during down-sampling or up-sampling), the shortcut connection is adapted using a convolution with a stride of 2 (He et al., 2016). This ensures that the dimensions of the input and output match. These are marked as dashed lines in Figure 3 of the original paper by He et al. (2016).

However, this breaks up the direct connection backbone that flows through the model and introduces additional complexity, as the convolution modifies the residual path and introduces non-linear transformations, disrupting the direct signal flow. The residual connections can be viewed from a different perspective entirely, as in Veit et al. (2016), where they act as the main pipeline. From this perspective, the convolutional blocks are ensemble shallow networks extracting and adding information to this main pipeline. Breaking up this pipeline with a convolution and activation makes this signal pipeline no longer linear.

The original ResNet design also applies a convolution between blocks with differing number of channels (He et al., 2016). However, this is not necessary for the CSLP-AE architecture where all blocks have the same number of filters.

Rather than using a convolution with stride 2 for the residual connection, interpolation can be used to resize the residual connection to match the target feature map dimensions. This approach retains the simplicity of identity mappings and the direct linear connection while also minimizing additional computational overhead.

Given the one-dimensional nature of the data, the choice of interpolation method is simply the piecewise linear interpolation of the signal to match the size of the down- or up-sampled signal in the time dimension. During down-sampling, the main convolutional layer reduces the feature map resolution, while the residual path is resized using linear interpolation to match the new resolution. Similarly, during up-sampling, the

residual is resized to match the expanded dimensions of the feature map, ensuring compatibility for the element-wise addition.

Let the input feature map be defined as $\boldsymbol{Z} \in \mathbb{R}^{C \times T_Z}$ where $C$ is the number of channels and $T_Z$ is the size of the time dimension, and let the output feature map after the convolution and activation be $\boldsymbol{U} \in \mathbb{R}^{C \times T_U}$ where $T_U$ is different from $T_Z$. The residual connection is interpolated to match the time resolution $T_U$ of the output as follows:

$$\boldsymbol{V} = \mathrm{interp}(\boldsymbol{Z}, T_U) + \mathrm{act}(\mathrm{conv}(\boldsymbol{Z})) = \mathrm{interp}(\boldsymbol{Z}, T_U) + \boldsymbol{U} \tag{20}$$

where $\mathrm{conv}(\boldsymbol{Z})$ is the convolution operation applied to the input $\boldsymbol{Z}$, $\mathrm{act}(\cdot)$ is the non-linear activation function, $\mathrm{interp}(\boldsymbol{Z}, T_U)$ is an interpolation function that resizes $\boldsymbol{Z}$ along the time dimension to match $T_U$, and $+$ functions as element-wise addition of the interpolated residual and the output of the convolutional block.

## H  Robust averaging procedures

### H.1  Tanh Weighting

tanh weighting scheme by Leonowicz et al. (2005) The tanh weighting scheme, introduced as a robust location estimator in Leonowicz et al. (2005), is applied across trials to improve ERP measurement by adaptively reducing the influence of outliers and non-stationary noise in the data. Unlike simple averaging, which treats all trials equally, the tanh method assigns weights based on the extremity of trial values (the amplitude), using a hyperbolic tangent function to down-weigh extreme deviations. In cases with a small number of trials, this scheme provides a significant advantage by dynamically optimizing weights in a data-dependent manner. This enhances the signal-to-noise ratio (SNR) and yields a more accurate and representative ERP (Leonowicz et al., 2005).

Denote a bootstrap sample of EEG signals in the most prominent channel as $\mathbf{X} \in \mathbb{R}^{K \times T}$ consisting of $K$ trials and $T$ timepoints. The weighting scheme is applied time-wise to estimate the best mean (location estimate) at each timepoint. First, the data for each time point $t$ is sorted in ascending order, producing a permutation $\rho_t(i)$ that maps indices from the original order to the sorted order as follows

$$x_{\rho_t(1),t} \leq x_{\rho_t(2),t} \leq \cdots \leq x_{\rho_t(K),t} \quad \forall t \in \{1, \ldots, T\} \tag{21}$$

$\rho_t(i)$ is the index in the original order that corresponds to the $i$-th smallest value in the sorted order. Specifically, $\rho_i : \{1, \ldots, K\} \to \{1, \ldots, K\}$ such that $x_{\rho_t(i),t}$ is the $i$'th smallest value of $\mathbf{X}_t$. The columns of the data matrix $\mathbf{X}$ are thus sorted in ascending order at each timepoint. After sorting, the rows no longer represent original trial indices but instead correspond to low-to-high voltage measurements. The weighting scheme is applied as follows

$$\hat{x}_t = \sum_{i=1}^{K} w_{i,t} x_{\rho_t(i),t} \quad \forall t \in \{1, \ldots, T\} \tag{22}$$

where $\boldsymbol{w}_t$ are weights per trial at time $t$. Applying a piecewise tanh-function to the new indices enables assigning lower weights to extreme voltages and higher weights to central values, improving the robustness of the location estimate. Such a function can be defined as follows

$$\kappa_{i,t} = \begin{cases} \tanh(c(i+1)) - v, & \text{if } i < \frac{K}{2}, \\ -\tanh(c(i-K)) + v, & \text{if } i \geq \frac{K}{2}. \end{cases} \quad \forall i \in \{1, \ldots, K\}, \forall t \in \{1, \ldots, T\} \tag{23}$$

$v > 0$ is a constant offset and $c > 0$ is a scaling parameter. The constant offset value is used to remove extreme values entirely from the equation by setting any negative weights to zero. The scaling parameter controls the curvature of the weighting function. For high values of $c$ the weighting approaches the uniform simple average and for lower values it approaches a linear scaling as a distance from the middle point. We use the default parameters from the original paper ($c = 0.1$, $v = 0$). Finally, the weights are obtained by normalizing the matrix such that each column vector sums to one:

$$w_{i,t} = \frac{\max(\kappa_{i,t}, 0)}{\sum_{j=1}^{K} \max(\kappa_{j,t}, 0)} \tag{24}$$

## H.2 Dynamic Time Warped Averaging

Dynamic Time Warping (DTW) for ERP estimation, as introduced in Molina et al. (2024), is a technique adapted from speech and sound processing (Berndt & Clifford, 1994; Müller, 2007) to address latency and jitter variability in EEG signals. Variations in trial timing and amplitude can distort averaged ERP waveforms, leading to blurred peaks and high ERP waveform variability (Ouyang et al., 2016; Murray et al., 2019). DTW aligns individual trials to a reference signal by minimizing temporal differences, improving ERP quality by reducing latency variability. This method is particularly useful for cases where latency jitter and amplitude variability across trials make simple averaging suboptimal.

Let $\mathbf{X} \in \mathbb{R}^{K \times T}$ denote a bootstrap sample of EEG signals for a specific channel, where $K$ is the number of trials and $T$ is the number of timepoints. Let $\boldsymbol{v} \in \mathbb{R}^T$ represent the reference signal, which is obtained using traditional simple averaging:

$$\boldsymbol{v} = \frac{1}{K} \sum_{k=1}^{K} \boldsymbol{x}_k \tag{25}$$

The DTW process dynamically adjusts each trial $\boldsymbol{x}_k \in \mathbb{R}^T, (k = 1, \ldots, K)$ by finding an alignment path $\boldsymbol{p} = (p_1, p_2, \ldots, p_M)$ where $p_m = (p_{i_m}, p_{j_m})$ maps the indices of $\boldsymbol{v}$ and $\boldsymbol{x}_k$.

The alignment path minimizes a total cost function defined by the sum of distances between the aligned elements:

$$c_k(i, j) = d(v_i, x_{k,j}) \quad \forall i, j \in \{1, \ldots, T\} \tag{26}$$

where $d(\cdot, \cdot)$ can be any distance function. In the present work the Euclidean distance $d(a, b) = (a - b)^2$ is used while in Molina et al. (2024) the Manhattan distance was used $d(a, b) = |a - b|$. The path $\boldsymbol{p}$ must satisfy the following two conditions:

1. The boundary condition $p_1 = (1, 1)$ and $p_M = (T, T)$.

2. Steps must be limited to $(1, 1)$, $(1, 0)$, or $(0, 1)$.

The first condition guarantees that the total length of both signals is considered avoiding partial matching. The second condition ensures that the path only moves in the down-right direction and prevents backtracking. These conditions ensure that the path starts in the upper-left corner and flows to the lower-left corner of the cost matrix.

The alignment path is found using dynamic programming (Senin, 2008). Using the alignment path $\boldsymbol{p}$ a warped version of each trial, $\boldsymbol{x}_k^w \in \mathbb{R}^M$, can be constructed. However, the signal has length $M$ which does not correspond with the length of each trial which is $T$. To ensure the warped trial has the same length as $\boldsymbol{v}$, steps in the alignment path $\boldsymbol{p}$ that do not advance the index of the reference signal (i.e. $(0, 1)$ steps) are discarded. The warped signal with these path elements discarded is denoted as $\boldsymbol{x}_k^{w*} \in \mathbb{R}^T$. Essentially, this looks at each index in the reference signal and takes the corresponding value of the trial of the path. In cases where there are multiple steps for the same reference index, Molina et al. (2024) opts to discard these. Another solution would be to take the average. In the present work these steps are discarded as well.

The estimated ERP is found by taking the average of the warped trials:

$$\boldsymbol{v}^* = \frac{1}{K} \sum_{k=1}^{K} \boldsymbol{x}_k^{w*} \tag{27}$$

By first matching and warping the signals to a common reference signal the temporal precision and amplitude consistency of the ERP is greatly improved and the influence of latency and jitter variability is reduced.

## H.3 Woody's Algorithm

Woody's algorithm (Woody, 1967) was implemented as a latency correction baseline. For each ERP component, we used the ERP CORE-defined measurement windows expanded in both directions to allow for broader alignment. The windows used are defined in Table 8. The algorithm iteratively aligned each trial to the evolving average until convergence, maximizing cross-correlation within the specified latency window.

Table 8: Woody Component Windows

| Component | Window (ms) |
|---|---|
| N170 | −110 to 350 |
| MMN | −125 to 425 |
| N2pc | 0 to 475 |
| N400 | 100 to 700 |
| P3 | 100 to 800 |
| LRP | −300 to 200 |
| ERN | −200 to 300 |

### H.4  RIDE

We implemented the Residue Iteration Decomposition (RIDE) method (Ouyang et al., 2014) in Python based on the publicly available MATLAB toolbox[2]. RIDE separates EEG signals into separate components: stimulus-locked (S), central (C), and response-locked (R) components. We used a three-component model (S, C, R) for all conditions except LRP and ERN, which were modeled with an R-only component. Window definitions followed the RIDE base guide, with main component windows set according to ERP CORE's reported significance intervals, see Table 9.

Table 9: RIDE Component Settings

| Component | S window (ms) | C window (ms) | R window (ms) |
|---|---|---|---|
| N170 | 110 to 150 | 100 to 900 | −300 to 300 |
| MMN | 125 to 225 | 100 to 900 | −300 to 300 |
| N2pc | 200 to 275 | 100 to 900 | −300 to 300 |
| N400 | 0 to 500 | 300 to 500 | −300 to 300 |
| P3 | 0 to 500 | 300 to 600 | −300 to 300 |
| LRP | – | – | −100 to 0 |
| ERN | – | – | 0 to 100 |

### H.5  Implementation validation

To verify that our Python implementation of the Woody latency correction baseline (Woody, 1967) and the RIDE decomposition baseline (Ouyang et al., 2014) behaved as expected, we validated both implementations on simulated ERP data with known single-trial latency variability. The simulated dataset consisted of 50 single-channel trials sampled at 500 Hz from −200 to 800 ms. Each trial contained two components with realistic morphology: a positive P300-like peak and a negative N200-like peak that preceded the P300 by 100 ms. The P300 latency varied uniformly between 300 and 500 ms across trials, and moderate additive noise, amplitude variation, and baseline drift were included.

Both methods estimated per-trial latency shifts, allowing direct comparison to the known ground-truth latencies. We quantified accuracy using the correlation between true and estimated latencies and the root mean square error after centering. Woody's algorithm recovered latencies with correlation 0.995 and RMSE 5.77 ms. RIDE achieved correlation 0.998 and RMSE 4.93 ms.

Figure 12 illustrates the unaligned and aligned trials, the recovered templates, and the correspondence between true and estimated latencies for both methods.

---

[2]A toolbox for residue iteration decomposition (RIDE): `https://cns.hkbu.edu.hk/RIDE.htm`

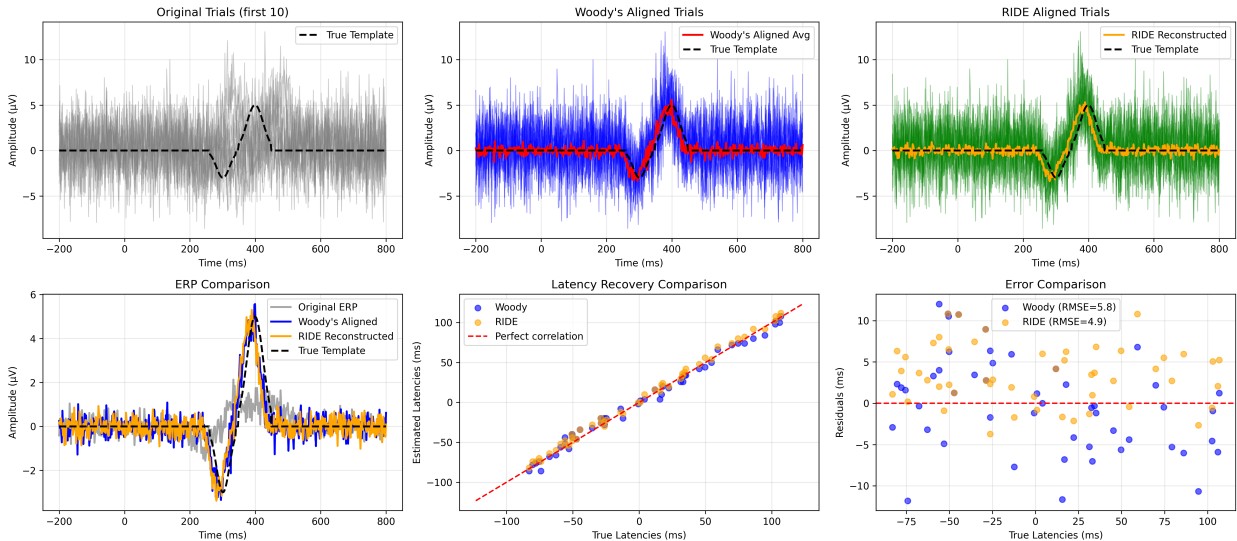

Figure 12: Validation of Woody's algorithm and RIDE on simulated ERP data with known latency jitter. Top panels show trial waveforms before alignment, after alignment with Woody, and after alignment with RIDE, together with the corresponding averages. Bottom panels show the averaged waveforms compared to the ground-truth template, the relationship between true and estimated latencies, and the residual estimation error. Both methods recover single-trial latency variability with high accuracy.

## I Experimental setup

We train EEGERP for $E_{\max} = 200$ epochs, annealing the predicted standard deviation over the first $E_{\text{target}} = 100$ epochs, and employ a learning rate of $4 \times 10^{-4}$ with a OneCycle learning-rate scheduler. Complete parameter values and experimental settings are detailed in the associated source code, which is provided as supplementary material and hosted at `https://github.com/andersxa/EEG2ERP`.

### I.1 Data partitioning

We evaluated EEG2ERP on three publicly available M/EEG datasets. For each dataset, subjects were partitioned into mutually exclusive training, validation (development), and test splits, and all experiments for that dataset used the same split. Table 10 summarizes the exact subject IDs per split.

Table 10: Data partitioning for all datasets.

| Dataset | Set | # subjects | Subject IDs |
|---|---|---|---|
| ERP CORE (40) | Training | 28 | 1, 2, 3, 6, 8, 9, 10, 11, 12, 13, 16, 17, 18, 19, 21, 24, 25, 28, 30, 31, 32, 34, 35, 36, 37, 38, 39, 40 |
| | Validation | 4 | 4, 7, 27, 33 |
| | Test | 8 | 5, 14, 15, 20, 22, 23, 26, 29 |
| Wakeman–Henson (16) | Training | 11 | 1, 2, 3, 5, 7, 8, 9, 10, 11, 13, 14 |
| | Validation | 2 | 12, 15 |
| | Test | 3 | 0, 4, 6 |
| P300 BCI Speller (55) | Training | 38 | 0, 1, 3, 4, 5, 6, 7, 8, 9, 10, 11, 13, 14, 15, 17, 18, 20, 21, 23, 27, 28, 34, 35, 36, 37, 38, 39, 41, 42, 44, 45, 46, 47, 49, 50, 51, 52, 53 |
| | Validation | 5 | 12, 29, 33, 40, 43 |
| | Test | 12 | 2, 16, 19, 22, 24, 25, 26, 30, 31, 32, 48, 54 |

## I.2 M/EEG data preprocessing pipeline

All M/EEG recordings were preprocessed in FieldTrip (Oostenveld et al., 2011). EEG and MEG were handled separately, following the steps summarized in Table 11.

Table 11: M/EEG preprocessing steps.

| Step | Operation | Details |
|---|---|---|
| 1 | Bad channel detection & interpolation | Channels with relative bandpower 49.75–50.25 Hz $> 15\%$ were marked as bad and interpolated from spatially neighboring channels. |
| 2 | High-pass filtering | FIR high-pass filter, 1 Hz cutoff, to remove baseline drifting. |
| 3 | Notch filtering | 50 Hz notch to remove power-line interference. |
| 4 | Segmentation (epoching) | Continuous data segmented into trials time-locked to task stimuli. |
| 5 | Artifact trial rejection | (i) reject trials with trial-level $z$-score$> 20$ (jump artifacts); (ii) reject trials with EOG channels band-pass 2–15 Hz and $z$-score$> 6$. |
| 6 | Baseline correction | Subtract mean in $-100$ ms to 0 ms pre-stimulus window. |
| 7 | EEG re-referencing | EEG only: average reference across EEG channels. |
| 8 | Resampling | Resampled to 200 Hz. |
| 9 | Trial cropping | Keep $-100$ ms to 800 ms per trial. |

