# OpenReview forum: "Estimating the Event-Related Potential from Few EEG Trials"
_TMLR — Accepted by TMLR_

### Review · Reviewer_mTSH · 2025-07-20

**Summary Of Contributions:**

This paper introduces a novel deep learning framework to map a few EEG trials to a denoised ERP waveform with uncertainty prediction. The main architecture is the existing CSLP-AE. The authors propose a bootstrap training scheme, a variance decoder, and new losses for the EEG2ERP framework. Experiments are performed on three public datasets and the model shows large gains over simple averaging, robust averaging and DTW, on unseen subjects. Additional analysis including ablation on the “averaging” procedure, visualization and analysis of the latent space, detailed evaluation of the output ERP features, are performed to enhance the rigorousness of the study. In sum, this is well-written, solid work with huge potential for various downstream applications and contributions to the neuroscience community.

**Audience:**

Yes

**Claims And Evidence:**

Yes

**Requested Changes:**

1. The ablation study is not complete, and the paper should ablate all differences vs CSLP-AE, which means studies of EEG2ERP w/o uncertainty, EEG2ERP w/o uncertainty w/o trial number embedding, and CSLP-AE with few trial averaged input should be included.
2. Technically speaking, we will not be able to get the “ground truth ERP” in real life experiments. Although the author designed a rigorous experiment paradigm, it would make the paper stronger if a simulation study can be performed to quantitively evaluate the model’s ability to recover ERP components (Latency, amplitude, topological map, etc).
3. I am particularly interested in the topography map in the real data as well as it also shows the spatiotemporal modeling abilities.
4. As shown in table 4-5, the latency metrics are not improving vs benchmarks but the amplitude estimates has a huge improvement. A discussion could be added on this point.

**Strengths And Weaknesses:**

Strength: The writing is clear. The study design is solid. The topic is important and has practical applications.

Weakness: Some of the experimental design can be further improved.

---

> ### Author Response · Authors · 2025-09-12
>
> We sincerely thank the reviewer for their thoughtful summary of our contributions and for highlighting the strengths of our work, including the clarity of writing, the solid study design, and the practical importance of the topic.
>
> **On the completeness of the ablation study:**
> We thank the reviewer for this suggestion. As noted in Appendix C, we have already included ablations comparing (i) CSLP-AE trained directly on single-trial inputs, and (ii) EEG2ERP operating on single-trial inputs ($K=1$), which demonstrates the role of our proposed procedure relative to prior approaches.
>
> We agree with the reviewer that additional ablations would further strengthen the study. In particular, we will include results for EEG2ERP without uncertainty modeling, EEG2ERP without uncertainty and without trial number embedding, and CSLP-AE using few-trial averaged input. These additional comparisons will provide a more complete picture of how each component of EEG2ERP contributes to performance.
>
> **On the simulation study:**
> We did not include a simulation study as the methods of choice here would likely depend on the design of the simulated data.
> That is to say, simulated data based on time-warped responses would favor the DTW procedure, data with simple outliers would potentially favor simple robust averaging procedures designed to handle such outliers, whereas highly non-linear variabilities could easily favor our approach.
> Consequently, instead of relying on simulation studies that potentially can be flawed by favoring some procedures over others due to the designed trial variabilities, we opted for evaluating the procedures on a variety of real datasets, that reflect realistic trial diversities.
> We then systematically compared the different methodologies in their ability to recover from few trials the ERP estimates obtained from a large number of trials.
>
> **On topography maps:**
> We will provide topography maps over time for different trial levels across the EEG2ERP model and the simple average baseline.
>
> **On latency vs. amplitude metrics:**
> We thank the reviewer for highlighting this point. As noted by the reviewer and shown in Tables 4-5, our approach yields improvements in amplitude estimation but does not show the same level of improvement for latency metrics compared to the benchmarks.
>
> This is consistent with the fact that our model still operates on averaged trial inputs, which makes it less robust to latency variability due to averaging over multiple trials perhaps with different latencies. An interesting direction is that, instead of simple averaging, our model could take as input the output of existing averaging and alignment baselines (e.g., DTW), thereby leveraging the strengths of prior state-of-the-art methods to precondition our model.
>
> To further improve robustness against non-stationarity, one could also extend the architecture to use separate encoders for each input trial or explicitly estimate and condition on latency. We will add this discussion to the revised manuscript and highlight it as an important avenue for future research.

---

> ### Author Response · Authors · 2025-09-24
>
> ## RE: More model ablations
> We extended the ablation study to include all conditions requested by the reviewer. In addition to the main EEG2ERP model, we evaluated EEG2ERP without uncertainty estimation, EEG2ERP without uncertainty and without trial number embedding, and CSLP-AE with few trial averaged input. All models were trained and evaluated under the same settings as the main experiments.
>
> | Gr.                          | n  | K=5           | 10%           | 100%          |
> |------------------------------|----|---------------|---------------|---------------|
> | CSLP-AE                      |    | -37.3         | -25.0         | -8.6          |
> | CSLP w/ Avg. Input           | 4  | -124.4±0.7    | -119.0±0.8    | -99.8±0.8     |
> | EEG2ERP w/ Single Trial      | 6  | 32.6±0.9      | 33.5±0.7      | 37.6±0.7      |
> | EEG2ERP                      | 10 | 31.7±0.6      | 36.1±0.5      | 47.7±0.4      |
> | EEG2ERP w/o Unc.             | 5  | 9.3±2.2       | 24.0±2.9      | 38.5±2.5      |
> | EEG2ERP w/o Unc. w/o Emb.    | 5  | 10.8±1.9      | 24.5±1.9      | 41.2±1.5      |
>
> The results show that removing uncertainty estimation reduces performance, particularly at lower data regimes. Removing both uncertainty and the trial number embedding actually increases reconstruction performance without uncertainty. Both models remain above the CSLP baselines and below the EEG2ERP model. CSLP-AE with few trial averaged input performs very poorly, yielding large negative R squared values across conditions. Together, these results confirm that uncertainty estimation contributes to the performance of EEG2ERP, and that the gap to CSLP-AE is not explained by trial averaging.
>
> ## RE: Topography maps
> We provide here the topography maps for subject 22 (test partition, so unseen during training) for N170/Faces. Showing the EEG2ERP prediction at different levels of trials and at peaks in the signal as determined by the ERP of the subject.
>
> The subject ERP is provided here, showing the time points of the topography map: https://i.imgur.com/aBh2SQE.png
>
> All EEG trials in a single plot is provided here: https://i.imgur.com/XegWeX9.png
>
> Finally, the topography map is provided here: https://i.imgur.com/eZs1lp5.png (Please revisit this plot since there was an error previously)
>
> We will provide more examples of these in the revised manuscript and discuss the spatiotemporal modeling abilities.

---

### Review · Reviewer_BojW · 2025-07-21

**Summary Of Contributions:**

The paper introduces a methodology for predicting the EEG event-related potentials (ERP) in the latter half of trials from a few initial trials. To do this it adapts an auto-encoding network trained to disentangle the subject and task specific information to handle the case of ERP input and outputs. In addition to the mean, the variance is also estimated and a negative log likelihood is used in lieu of mean squared error for the reconstruction. Results show that with just 1 or 5 trials the model often has positive R^2 for predicting the average ERP, which is much better than baselines.

**Audience:**

Yes

**Broader Impact Concerns:**

The broader impact statement appears to be too broad for this scope of this study.

**Claims And Evidence:**

No

**Requested Changes:**

The paper should be revised to address weaknesses above (critical).

The phrasing " it also provides variance estimates that are learned during training using associated bootstrapped ERP training targets accounting for the ERP uncertainty." is hard to parse. The confusion is that the uncertainty estimates use a Gaussian likelihood model, which is independent of the bootstrapping. The way it was written I was anticipating that the bootstrap variance estimates were going to be used.

Last sentence before Figure 1 "ERPs to quantify reconstruction accuracy." I'm not sure that the choice of 'reconstruction' should be used. It is actually misleading because the target is not reconstructed per sea. It is rather a prediction problem given the chronological split between initial and final trials.


**Issues with formulation (critical)**:
"To support efficient training, we construct a batch containing all combinations of subject-task pairs." Taken literally, this doesn't seem possible for a large number of subjects and tasks. The aggregating in (9) and (10) means subjects are paired and then tasks are paired. Is that correct, rather than all combinations. Also, the verbal description before and after equation 9 and 10 is not clear. As I think 'non-corresponding axis' should be corresponding axis, i.e., same task in (9) or same subject in (10). Also do $A$ and $B$ denote disjoint set of $N_S$ subjects? Otherwise, why not use the same tensor for both? Furthermore, there must be a mistake in indexing because $j$ is undefined in (9) and $i$ is undefined in (10). Also the term "opposing axis" isn't clear.

It is rather confusing, what the pairs are in relation to $\bar{\mathbf{Z}}^{A,\mathbb{L}}$ compared to without the bar defined before equation 9. The operation of the random permutation is not clear in (11), I guess $\bar{\mathbf{X}}$ denotes a single realization case, but it is not clear what it corresponds to given that $A$ and $B$ are different tensors. Also it is not clear if there should a sum over it. That is, is this is an instance wise loss like (7)? Or a batchwise loss like (8)?

What is the 'pre-latent embedding space'?

**Additional results**: (possibility to strengthen the work)
I'd request to compare the linear annealing of predicted scale with the Beta-NLL approach for learning the uncertainty estimate. Also the uncertainty estimate can be quantified using expected calibration error (i.e., binning the variance estimates and looking at the squared error compared to the bin-center variance for points assigned to each variance bin).

A simple baseline would be to find the nearest neighbor (considering ERPs from full set of trials) of the training set, and retrieve that. This could be more meaningful than global average template.

I'd also like to see Woody's jitter based approach.

There is description in Appendix E and Appendix F about the changes from the CSLP-AE baseline.  The additional modifications of using gated linear units is quite substantial, but it is not clear the impact of these choices compared to overall formulation of the problem.

The results are impressive for K=1 and K=5, but its not clear to me if that is the first and first 5 trials respectively, or it if is across all sets of 1 or 5 trials. But in Table 2, at 100% the results are well below simple averaging. Perhaps the Zipf law on the number of trials overemphasizes learning from a few trials at the expense of performing well when many trials are given. This is related to the issue of noise scheduling in denoising auto-encoders that has lead in part to  diffusion models. There is no simple answer but it seems that different priors on $K_b$ could be compared.

Similarly, given the training over different numbers of trials in (2), wouldn't it be better to test? Also I don't see how information leakage during training is a problem as long as the splits are chronological. Meaning that the if the input average is based on a boostrap average of the first K trials, couldn't the remaining N-K trials be used?


**Questions**

It would be useful to describe the baseline methods with more detail. Is the "global template" the average across other subjects?  The exact phrase only appears in the tables.

"EEG2ERP is the first method to map EEG signals directly to their associated ERPs" is this correct? Besides $K=1$ isn't it operating on averages also?

How are the tanh parameters optimized? Original paper uses Nelder-Mead. Is the same adopted? Nothing is mentioned in Appendix G.

In tables 1 and 2, for $K=1$ the entry for Weighted should be dashed out, as there is no sense in this.  Also what is the sense of DTW for $K=1$? That is DTW warping to itself? I feel that DTW with its own reference should also be grayed out.

Why isn't the Weighted (Leonowicz et al. 2005) included in Table 3.


**Minor notes:**

Figure 1, internal notation of superscripts with parentheses and subscripts b and n do not match the equations.

Using calligraphic D for both the dataset and the decoder can be confusing. I'd avoid calligraphic for the encoder and decoder.

Last sentence before Section 3. "Hyperparemeters" misspelled.

Why would xDAWN "acts on two or more components reducing the SNR"? I think this is a typo.

I find it odd that Table 1 appears at top of page in the midst of Section 4 well before its reference in Section 5.

**Strengths And Weaknesses:**

**Strength:** The approach of performing non-linear denoising/prediction from bootstrapped average ERPs is interesting. It could connect to wider literature on denoising auto-encoders.

The included experiments and figures are clear and seem well defined.

**Weaknesses:**

The major gap is the lack of connection between the problem that is solved and the ultimate aim of reducing trials for a specific purpose. The approach aims to reduce number of trials to get reliable ERP estimates. But the ultimate aim, what can you do with a ERP estimate is under-explored, which undermines the paper.  The big question is whether the estimate of ERP with fewer samples sufficient information to act on. It is not clear if it help neuroscience studies or enhance the classification in brain-computer interfaces.

There is a need for a discussion regarding whether the few trial R^2 (possibly combined with the variance estimates) is sufficient to support scientific hypothesis testing.

Note that the Molina et al. paper uses classification in the end to justify their approach. Also xDAWN also examines the accuracy of classification as a function of the number of repetitions. For the P300 speller, the problem is to classify the ERP as being target or non-target. It's not clear if the improved reconstruction in low trial regime supports that. "The ability to estimate ERPs from very few trials opens the door to faster, more efficient EEG-based studies." This is a possible implication but it should be justified.

It should be made more clear early on that the ERP estimate from the many trials (second half of the trials) is considered the gold standard / ground truth.  However, this assumption is not clear given the fact that there could be temporal jitter and/or artifacts. I.e. for training and testing different versions of what is considered ground truth ERP could be used.

Regarding jitter, a classic baseline for ERP estimation with misaligned trials is Woody, Charles D. "Characterization of an adaptive filter for the analysis of variable latency neuroelectric signals." Medical and Biological Engineering 5, no. 6 (1967): 539-554. This is arguably simpler than the dynamic time warping.

A base assumption is that the EEG trials are partitioned into two halves... I assume these halves are chronological. It is idealistic to consider that there is a single 'ERP' as the brain adapts. Habituation can happened after only a few trials.  Early and late trials may be distinct, which is in itself interesting. This is not discussed.  Looking at the results, could the ERP "ERN/Correct" in Subject 20, be an example of habituation?

Clarity is also issue at times with the formulation, there are some mistakes and ambiguities.

Another weakness is the heuristic choice of annealing the variance estimate in the Gaussian negative log-likelihood as this specific problem has been addressed in the literature (Beta-NLL paper):
Seitzer, Maximilian, Arash Tavakoli, Dimitrije Antic, and Georg Martius. "On the Pitfalls of Heteroscedastic Uncertainty Estimation with Probabilistic Neural Networks." In Tenth International Conference on Learning Representations (ICLR 2022). 2022.

---

> ### Author Response · Authors · 2025-09-12
>
> We thank the reviewer for their careful and constructive feedback. We will clarify the noted misunderstandings and address the suggested points to improve the manuscript.
>
> ## RE: The Ultimate Aim of Reducing Trials
> Our objective is not to improve a downstream classifier but to recover interpretable, component-level event-related potentials that scientists routinely analyze to test hypotheses about cognition, perception, and disease. These exploratory studies depend critically on the quality of the ERP waveform: its amplitude and morphology must be cleanly measurable to serve as reliable dependent variables. For example, Kutas and Federmeier (2011) emphasize the utility of the N400 amplitude in indexing semantic memory retrieval and integration, Brouwer and Hoeks (2013) highlight how N400 and P600 amplitudes map onto distinct cognitive processes within a language comprehension network, and Fields (2023) reviews the late positive potential (LPP) predominantly in the context of its amplitude modulations linked to emotional processing and memory. Reliable access to these signal-level measures with fewer trials therefore directly benefits exploratory and analytic neuroscience.
>
> Our framework is directly aligned with best practices outlined for ERP research in clinical populations. As noted by Luck and Kappenman (2016), ERP studies in patient groups face the dual challenge of limited trial numbers and the need for highly reliable measures. ERPs are only useful as biomarkers or dependent measures when their amplitudes and morphologies can be cleanly quantified, yet data constraints often compromise this. By improving ERP quality under reduced trial counts, our approach addresses exactly these concerns, thereby supporting exploratory neuroscience and clinical applications in line with established methodological recommendations.
>
> A further motivation for our work is the well-documented dependence of ERP reliability on the number of trials. For example, Huffmeijer et al. (2014) demonstrated adequate to excellent test-retest reliability for multiple ERP components only when sufficient trial numbers were available, recommending at least 30 trials for early components such as the VPP and more than 60 trials for later, broadly distributed components such as the P3. Similarly, developmental and lifespan studies have shown that error-related components such as the ERN and Pe (error positivity) can be reliably measured with as few as six to eight trials, but only under favorable conditions and with carefully controlled tasks (Pontifex et al., 2010; Meyer et al., 2014). Importantly, child ERP guidelines explicitly recommend minimizing trial numbers and building paradigms around the stability constraints of ERP components (Brooker et al., 2021). These converging findings illustrate that ERP reliability is strongly trial-limited and that many populations (children, clinical groups, older adults) can not realistically provide the large number of artifact-free trials typically required.
>
> Our framework therefore provides a methodological solution by improving ERP quality at lower trial counts, enabling robust measurement of components like the P3, LPP, and ERN even in contexts where collecting large numbers of repetitions is impractical.
>
> ### References
>
> Kutas, Marta, and Kara D. Federmeier. "Thirty years and counting: finding meaning in the N400 component of the event-related brain potential (ERP)." Annual review of psychology 62.1 (2011): 621-647.
>
> Brouwer, Harm, and John CJ Hoeks. "A time and place for language comprehension: mapping the N400 and the P600 to a minimal cortical network." Frontiers in human neuroscience 7 (2013): 758.
>
> Fields, Eric C. "The P300, the LPP, context updating, and memory: What is the functional significance of the emotion-related late positive potential?." International Journal of Psychophysiology 192 (2023): 43-52.
>
> Kappenman, Emily S., and Steven J. Luck. "Best practices for event-related potential research in clinical populations." Biological psychiatry: cognitive neuroscience and neuroimaging 1.2 (2016): 110-115.
>
> Huffmeijer, Renske, et al. "Reliability of event-related potentials: The influence of number of trials and electrodes." Physiology & behavior 130 (2014): 13-22.
>
> Pontifex, Matthew B., et al. "On the number of trials necessary for stabilization of error‐related brain activity across the life span." Psychophysiology 47.4 (2010): 767-773.
>
> Meyer, Alexandria, Jennifer N. Bress, and Greg Hajcak Proudfit. "Psychometric properties of the error‐related negativity in children and adolescents." Psychophysiology 51.7 (2014): 602-610.
>
> Brooker, Rebecca J., et al. "Conducting event-related potential (ERP) research with young children." Journal of psychophysiology (2019).

---

> > ### Author Response · Authors · 2025-09-12
> >
> > ## RE: Sufficient Performance Metrics
> > $R^2$ in our study is used to assess reconstruction quality, that is, how effectively the model reduces noise relative to a target ERP. While this provides a useful criterion for model selection, it is not intended as a direct metric for scientific hypothesis testing. What is essential for scientific use is the ability to quantify component-level amplitudes and latencies with sufficient reliability. This has been the standard in ERP methodology (e.g., Huffmeijer et al. 2014; Brooker et al. 2021), and our results show that our framework improves access to these measures under trial-limited conditions (see Tables 4 and 5 in the Appendix).
> >
> > Furthermore, we can generate samples (via simulation) from the predictive distribution defined by the model’s mean and variance estimates. For each sampled ERP, morphological metrics such as amplitude and latency can be computed, and aggregating across samples yields empirical distributions for these metrics. In some cases, these distributions can also be derived directly from the predictive distribution itself. This enables not only point estimates of ERP components, but also principled uncertainty quantification over amplitudes and latencies.
> >
> > ## RE: On Using Classification to Justify the Approach
> > Our primary objective is to enable reliable recovery of interpretable ERP components for hypothesis-driven neuroscience, rather than to optimize a downstream classifier. Nevertheless, we agree that classification performance is an important complementary indicator of whether improved reconstructions preserve discriminative information. To this end, we provide classification analyses in Appendix A (Figures 4 and 6). With K=1 single-trial inputs, balanced accuracies were 35.86% (subject), 22.30% (task), and 43.95% (ERP paradigm). With only K=5 trials, accuracies increased substantially to 68.49% (subject), 53.77% (task), and 78.34% (ERP paradigm). These results demonstrate that improved ERP quality at low trial counts does indeed support classification performance, consistent with prior work such as xDAWN and Molina et al. While classification is not our main evaluation criterion, these findings further support the implication that denoised low-trial ERPs open the door to faster and more efficient EEG-based studies.
> >
> > ## RE: Ground Truth ERP
> > We agree that an ERP obtained from many trials can not be considered a literal gold standard, as it remains an empirical estimate that is itself subject to jitter and artifacts. In our work, we use the many-trial average as a practical reference, consistent with established practice in ERP methodology where averaging over large trial counts is the standard approach to estimate the underlying component with a high enough trial count. Our framework is therefore evaluated against this high-SNR reference, not because it is the true ground truth, but because it provides a strong benchmark for assessing how well lower-trial estimates recover interpretable ERP components that we should be able to at least partially reproduce from another equally strong estimated ERP (from the first half). We will clarify this point earlier in the manuscript to avoid confusion.
> >
> > ## RE: Woody Baseline
> > We thank the reviewer for this suggestion and will include this baseline in the paper and provide results here in the coming days.
> >
> > ## RE: Order of EEG Trials
> > Our input/target split is drawn uniformly, not chronologically, so habituation effects should not affect the results as such effects would be uniformly distributed in each half. The case you mention (Subject 20, ERN/Correct) is more likely due to an outlier in the sampled $K=5$ trials, which skews the ERP estimate toward the end of the epoch. This is consistent with the higher uncertainty observed in the same region. We will make the random-split procedure more explicit in the paper.
> >
> > ## RE: Clarity and Ambiguities
> > We thank the reviewer for noting issues of clarity in the manuscript. We will carefully revise the text to improve precision and readability. It would also be very helpful if the reviewer could indicate specific instances where mistakes or ambiguities remain, so that we can address them directly in a revision.

---

> > > ### Author Response · Authors · 2025-09-12
> > >
> > > ## RE: Annealing Variance and Beta-NLL
> > > The $\beta$-NLL loss indeed provides a principled approach to mitigating instabilities in heteroscedastic training. Our variance annealing strategy is intended as a heuristic generalization of the staged optimization approach proposed in Stirn et al. (2023). In that framework, the model is first trained with squared error loss on the mean (equivalent to fixing the variance at one), and only afterwards is the variance estimator optimized using the negative log-likelihood.
> > >
> > > Our interpolation scheme replaces this discrete two-stage procedure with a smooth transition, linearly annealing the variance contribution from a fixed value of one toward the model's predicted variance over the course of training. This can be seen as a heuristic extension of the faithful training principle, providing a more gradual training signal that stabilizes training.
> > >
> > > Importantly, Stirn et al. (2023) compared their method against $\beta$-NLL and demonstrated that their optimization reliably achieves strong mean accuracy and variance calibration across a wide variety of tasks, in some cases outperforming $\beta$-NLL. Since our method directly generalizes this framework, we position our contribution in that lineage rather than in opposition to $\beta$-NLL. We will add this to the main manuscript and clarify our approach.
> > >
> > > ### References:
> > >
> > > Stirn, Andrew, et al. "Faithful heteroscedastic regression with neural networks." International Conference on Artificial Intelligence and Statistics. PMLR, 2023.
> > >
> > > ## RE: Learned Variance Estimates
> > > We thank the reviewer for pointing out this ambiguity. Our intention was not to suggest that bootstrap variance estimates are directly used in the likelihood. Rather, the bootstrap procedure is used to construct ERP training targets that naturally contain variability due to sampling. The model is then trained with a Gaussian likelihood, where the variance is parametrized and learned by the network itself. In other words, the variance estimator is not derived from the bootstrap but instead attempts to capture the intrinsic uncertainty present in the bootstrapped ERP targets. We have revised the phrasing in the manuscript to make this distinction clear.
> > >
> > > ## RE: Reconstruction vs. Prediction
> > > Standard autoencoder nomenclature uses reconstruction, which is where this term stems from. We agree, however, with the reviewer that prediction is a more precise term for this specific problem, and we will revise the manuscript to reflect this. We further note that the splits are not chronological in initial and final trials but random, as explained earlier in "RE: Order of EEG Trials".

---

> > > > ### Author Response · Authors · 2025-09-12
> > > >
> > > > ## RE: Batch Construction
> > > > We thank the reviewer for pointing out the ambiguities in our description of the contrastive loss.
> > > >
> > > > **On "all combinations of subject-task pairs":** This is indeed to be taken literally.
> > > > For $N_S$ subjects and $N_T$ tasks we construct a batch of size $N_S \times N_T$, which is feasible in practice
> > > > (e.g. 40 subjects and 14 tasks yield 560 pairs, which is within the typical range of batch sizes used for EEG models).
> > > > Each pair consists of two tensors, $\mathbf{Z}^A$ and $\mathbf{Z}^B$.
> > > > In situations where constructing all combinations is not feasible, it is equally valid to subsample random subject-task pairs, and our framework supports this option explicitly.
> > > > We will clarify this point in the manuscript.
> > > >
> > > > **On the role of $A$ and $B$:** Both $\mathbf{Z}^A$ and $\mathbf{Z}^B$ contain embeddings for the same set of subjects and tasks in the same order. They are not disjoint sets, but rather two random realizations of ERPs from each subject and task pair encoded into latents.
> > > > The contrastive loss is then applied between these two.
> > > >
> > > > **On aggregation and indexing: The reviewer is correct that our original notation was unclear.
> > > > When constructing similarity matrices, we aggregate over the axis that does not correspond to the latent space of interest.
> > > > For subject space, we compare embeddings across subjects while summing over tasks.
> > > > For task space, we compare embeddings across tasks while summing over subjects.
> > > > The corrected formulations are:
> > > >
> > > > $$
> > > > \mathbf{S}^{\mathbb{S}}\_{i j} = \sum\_{\tau=1}^{N_T}
> > > > \operatorname{sim}(\mathbf{Z}^{A,\mathbb{S}}\_{i,\tau},\mathbf{Z}^{B,\mathbb{S}}\_{j,\tau}),
> > > > \quad \forall i,j \in \{1,\ldots,N\_S\},
> > > > $$
> > > >
> > > > $$
> > > > \mathbf{S}^{\mathbb{T}}\_{i j} = \sum\_{s=1}^{N_S}
> > > > \operatorname{sim}(\mathbf{Z}^{A,\mathbb{T}}\_{s,i},\mathbf{Z}^{B,\mathbb{T}}\_{s,j}),
> > > > \quad \forall i,j \in \{1,\ldots,N\_T\}.
> > > > $$
> > > >
> > > > These yield an $N_S \times N_S$ similarity matrix in the subject space and an $N_T \times N_T$ similarity matrix in the task space, which are then used in the symmetric temperature-scaled cross-entropy loss.
> > > >
> > > > **On terminology:** We agree that "opposing axis" was unclear.
> > > > We have revised the text to instead state "we aggregate over the non-target axis," that is, over tasks when constructing subject similarities and over subjects when constructing task similarities.
> > > >
> > > > We have corrected the equations and clarified the description in the manuscript accordingly.
> > > >
> > > >
> > > > ## RE: Latent Permutation Loss
> > > > We thank the reviewer for pointing this out. We agree that the notation was confusing, and we will remove the bar notation to avoid ambiguity. The latent tensors $\mathbf{Z}^{A,\mathbb{L}}$ and $\mathbf{Z}^{B,\mathbb{L}}$ used in the latent permutation loss are the same tensors as in the contrastive loss. The bar notation was a remnant of an earlier draft. We will remove it in the revised manuscript.
> > > >
> > > > Regarding the interpretation of the loss: the latent permutation loss is indeed an instance-wise loss, analogous to the reconstruction loss in Equation~(7). Specifically, $\bar{\mathbf{X}}$ refers to a single ERP realization, and the decoder prediction is computed from the permuted latent tensors as
> > > >
> > > > $$
> > > > D\_\theta(\varrho\_2(\mathbf{Z}^{A,\mathbb{S}}), \varrho\_1(\mathbf{Z}^{A,\mathbb{T}})).
> > > > $$
> > > >
> > > > Here, the permutation operator $\varrho_i(\cdot)$ randomly reorders elements along the axis not associated with the current latent space. For example, when operating in the subject space $\mathbb{S}$, we apply $\varrho_2$, which permutes the task axis so that the $N_T$ tasks are randomly reordered for the subject latents. The reconstruction is then compared to the original ERP via the negative log-likelihood reconstruction loss just as in Equation~(7).
> > > >
> > > > We will revise the text to clarify that the latent permutation loss is computed on individual ERP instances in the same manner as Equation (7).
> > > >
> > > > ## RE: Pre-latent Embedding Space
> > > > The pre-latent embedding space in the CSLP-AE model denotes the latent space after the encoder, but before the bottleneck transformer. That is, the embedding is added in this space before the bottleneck transformer which outputs into the latent space. We will clarify this better in the revised manuscript.

---

> > > > > ### Author Response · Authors · 2025-09-12
> > > > >
> > > > > ## RE: Additional Results
> > > > > **On the comparison to $\beta$-NLL and calibration:**
> > > > > We thank the reviewer for this further suggestion. As outlined above, our method is a heuristic generalization of the faithful training procedure of Stirn et al. (2023), who directly compared their method against $\beta$-NLL (Seitzer et al., 2022) and in many cases demonstrated superior mean accuracy and variance calibration. For this reason, we consider our contribution as part of that lineage rather than requiring a separate $\beta$-NLL comparison.
> > > > >
> > > > > Regarding the evaluation of calibration using expected calibration error (ECE), we agree it could also be possible to display the calibration in terms of bins of the variance estimates and using center values. However, we believe the present representation of the calibration directly plotting estimated standard deviations to actual root-mean square reconstruction error is a more direct and intuitive approach to inspect how calibrated the uncertainty estimates are.
> > > > >
> > > > > **On the nearest-neighbor baseline:**
> > > > > We thank the reviewer for this helpful suggestion. In addition to the global average template baseline, we will include a
> > > > > nearest-neighbor baseline that retrieves ERPs from the training set. Specifically, we will gather per-subject ERPs across
> > > > > tasks and perform a task-conditioned nearest-neighbor search to select the most similar ERP as the prediction. This new
> > > > > baseline will be added to the revised manuscript for completeness.
> > > > >
> > > > > **On the use of gated linear units:**
> > > > > We agree with the reviewer that the replacement of the ReLU activation with gated linear units (GLUs) is a substantial architectural change relative to the CSLP-AE baseline. In the broader context of deep learning, this can be interpreted as updating the activation function to a more state-of-the-art choice. However, in the specific context of EEG and ERP data, GLUs are particularly well-suited for the denoising and reconstruction task.
> > > > >
> > > > > In the original CSLP-AE, ReLU was combined with instance normalization but without any residual pathway. As a result, important amplitude information was often suppressed, since instance normalization removes magnitude differences temporally across latent representations. This is problematic because amplitude and relative temporal magnitude differences are crucial for accurately reconstructing ERPs.
> > > > >
> > > > > GLUs address this limitation by combining a gating mechanism with a linear pathway. The gating mechanism provides nonlinear expressivity, while the linear pathway preserves amplitude information that would otherwise be lost. This enables the model to better retain and restore amplitude-related features during reconstruction. We have revised the manuscript to clarify the motivation for this modification and its relevance for ERP reconstruction. Importantly, we already provide a direct comparison between the original CSLP-AE and the upgraded EEG2ERP architecture in Table~6 of the Appendix.
> > > > >
> > > > > **On the role of trial scheduling and Zipf's law:**
> > > > > We thank the reviewer for this insightful observation. As described in Section 3.1, all metrics are computed using $B=200$ bootstrapped subsets of input trials, which ensures that results are not tied to a particular ordering of trials.
> > > > >
> > > > > Regarding the reviewer's comment on Zipf's law and noise scheduling: Equation (2) in our paper specifies how we sample the number of trials during training, which effectively acts as a scheduling distribution. In early development, we experimented with several alternative schedules (e.g. $1-k/K$, $1/k$, and other formulations), but empirically we found that the proportionally inverse schedule provided the best performance in the low-trial regime ($K=1,5$).
> > > > >
> > > > > This choice indeed represents a trade-off that emphasizes learning from few trials at the expense of optimal performance when many trials are available, reflecting the heavy-tailed nature of the Zipf distribution. While this prioritization was motivated by our target application setting, we agree that alternative priors or scheduling strategies could yield different trade-offs. We will clarify this discussion in the manuscript and highlight scheduling design as an important future research direction, analogous to the role of noise scheduling in denoising auto-encoders and diffusion models.

---

> > > > > > ### Author Response · Authors · 2025-09-12
> > > > > >
> > > > > > ## RE: Questions
> > > > > > **On the description of baseline methods:**
> > > > > > We thank the reviewer for this comment. We agree that the baselines require a more detailed description.
> > > > > > The "global template" baseline corresponds to averaging over all trials and all training subjects for each task, and then retrieving this per-task global ERP during evaluation. We will update the manuscript to include this clarification and ensure that the baselines are described consistently in both the main text and the tables.
> > > > > >
> > > > > > **On the phrasing "EEG2ERP is the first method to map EEG signals directly to their associated ERPs":**
> > > > > > We thank the reviewer for raising this point. To our knowledge, prior work on single-trial EEG has primarily focused on *quantification*, such as estimating component latencies, amplitudes, or performing classification, rather than reconstructing the entire ERP waveform. In contrast, EEG2ERP is designed to reconstruct the full ERP from fewer and fewer trials all the way down to single-trial EEG data.
> > > > > >
> > > > > > For $K=1$, this corresponds to mapping directly from a single-trial EEG signal to its associated ERP, which is the intended meaning behind our phrasing. For $K>1$, the model operates on the pointwise average of $K$ trials, which provides a denoised estimate of the ERP but still requires reconstructing the full ERP waveform. We will clarify this distinction in the manuscript to avoid ambiguity.
> > > > > >
> > > > > > **On the tanh baseline parameters:**
> > > > > > We follow the original paper and use the default parameters ($c=0.1, v=0$) without Nelder–Mead optimization on our dataset. We will make this clear in the revised Appendix G.
> > > > > >
> > > > > > **On baselines for $K=1$:**
> > > > > > We thank the reviewer for this observation. For $K=1$, the weighted averaging and dynamic time warping (DTW) baselines indeed reduce to degenerate cases: the weighted sum is identical to simple averaging, and DTW would simply align a trial with itself. While in principle one can still compute these, we agree that they are not meaningful as baselines in the single-trial case.
> > > > > > In the revised manuscript, we will put dashes in the entries for Weighted and DTW at $K=1$ in Tables 1 and 2, and retain only the simple averaging baseline, which is the natural comparison for a single trial.
> > > > > >
> > > > > > **On the Weighted (Leonowicz et al. 2005) exclusion from Table 3:**
> > > > > > This is an oversight on our part and we will add this baseline to the table as well.
> > > > > >
> > > > > > ## RE: Minor Notes
> > > > > > Thank you for the helpful notes. We have corrected: (i) Figure 1 notation (superscripts/subscripts), (ii) the encoder/decoder symbols (avoiding calligraphic $\mathcal{E},\mathcal{D}$), (iii) the typo "Hyperparameters", and (iv) the placement of Table 1. (v) we corrected the typo; xDAWN *increases* the SNR of the evoked response (not reduces).
> > > > > >
> > > > > > ## RE: Broader Impact Concerns
> > > > > > We agree with the reviewer's concern that the impact statement could be more specific. We will revise it to focus directly on the potential benefits and limitations of our EEG2ERP framework.

---

> > > ### Comment · Reviewer_BojW · 2025-09-26
> > > **concern about non-chronological split**
> > >
> > > I appreciate the authors thoughtful reply and new results. I have read them carefully. Sorry for missing Figure 8 in the Appendix which showed the decreases in task performance using single trial predictions. It seems to be a large drop in performance.
> > >
> > > A concern that I expressed in my original view was regarding the splits, and I made a few statements assuming chronological splits were used. When non-chronological splits are used (both for training and evaluation) how does this translate into the real-world practice that rebuttal further strengthened? I mean if the number of trials is to be reduced, wouldn't only the latter trials be left off?

---

> > > > ### Author Response · Authors · 2025-10-01
> > > >
> > > > We thank the reviewer for this important point. We agree that in practice the relevant use case is to reduce the number of trials by leaving out the later trials, which corresponds to using a sequential chronological ordering rather than random samples from all trials. To address this, we ran an additional experiment where all methods were evaluated under this setting. Specifically, we used EEG2ERP as trained on randomly ordered bootstrap samples as before, but evaluation was carried out on chronologically sequential trials.
> > > >
> > > > | Gr.              | n  | K=5          | 10%          | 100%      |
> > > > |------------------|----|--------------|--------------|-----------|
> > > > | EEG2ERP          | 9  | **32.7±2.7** | **43.7±0.8** | 50.4±0.5  |
> > > > | DTW    |    | -65.6        | -66.7        | 59.9      |
> > > > | Weighted |    | -147.9       | -135.0       | **65.3**  |
> > > > | RIDE             |    | -232.5       | -133.5       | -16.7     |
> > > > | Woody's alg.     |    | -226.3       | -209.1       | 18.6      |
> > > > | Nearest neighbor |    | 25.7         | 13.3         | 47.7      |
> > > > | Simple avg.      |    | -171.6       | -159.0       | 60.7      |
> > > >
> > > > These results show improved performance compared to random sampling. This is expected, since temporally adjacent trials are less affected by variability such as changes in fatigue or other brain states. Nevertheless, the key observation is that EEG2ERP continues to perform strongly under this practically relevant sequential evaluation.

---

> > > > > ### Comment · Reviewer_BojW · 2025-10-02
> > > > > **an important step towards the practical utility**
> > > > >
> > > > > Thank you for this result!
> > > > >
> > > > > First, training with chronological splits is likely to do better and also be interesting in that it could account for habituation effects.  Agree?
> > > > >
> > > > > Secondly, practically, the estimate of the chronological split prediction and the observed trials (pre-split) could together be used (perhaps with a weighting scheme) to produce a single ERP estimate.  I understand that for evaluation purposes the independence between input set of trails and match is desired, but in the ideal use case, one would use the combined estimate (a function of the early trials that are used to make the prediction and the prediction itself) and compare that to the average across all trials. This would give confidence that only collecting a subset of early trials and using those to form an estimate of hypothetical later trials would give a better estimate of the ERP for all trials (collected and hypothetical).  Thus, I encourage the authors to discuss this.

---

> > > > > > ### Author Response · Authors · 2025-10-06
> > > > > >
> > > > > > We thank the reviewer for these constructive suggestions. We agree that training with chronological splits is likely to yield further improvements, as it would allow the model to directly capture habituation effects and other temporal dependencies in the data.
> > > > > >
> > > > > > Regarding the idea of combining observed early-trial averages with model-predicted later trials, we see the appeal of this approach. In practice, one would not discard the observed early-trial ERP but could instead integrate it with the model's predictions of the later trials to form a combined estimate. At the same time, we note that in the very low-trial regime that motivates our work, the simple average is highly noisy and contributes little on its own, which limits the immediate benefit of such a combination. Nevertheless, for higher trial counts or related settings, this strategy could indeed be useful.
> > > > > >
> > > > > > More broadly, we emphasize that our framework is flexible in the sense that it can accept inputs of varying signal-to-noise ratios. This means that in practice, previously obtained ERP estimates, whether from simple averages, established denoising baselines, or even the model's own prior outputs, could be provided as inputs alongside new trials to produce an even cleaner ERP estimate. We believe this kind of iterative refinement and integration of multiple sources of information aligns well with the reviewer's suggestion and represents an important direction for future work.

---

> ### Author Response · Authors · 2025-09-24
>
> ## RE: Additional baselines
> We implemented Woody's algorithm (Woody 1967), RIDE (Ouyang et al., 2015), and Nearest Neighbor baselines. For RIDE, we used a three component model with stimulus locked (S), central (C), and response locked (R) components, except for LRP and ERN which were modeled with only an R component. The windows were chosen following the RIDE guide for base windows, with the main component window set according to the significant windows reported in ERP CORE. Specifically:
>
> - **N170**: S = 110–150 ms, C = 100–900 ms, R = -300–300 ms
> - **MMN**: S = 125–225 ms, C = 100–900 ms, R = -300–300 ms
> - **N2pc**: S = 200–275 ms, C = 100–900 ms, R = -300–300 ms
> - **N400**: S = 0–500 ms, C = 300–500 ms, R = -300–300 ms
> - **P3**: S = 0–500 ms, C = 300–600 ms, R = -300–300 ms
> - **LRP**: R = -100–0 ms
> - **ERN**: R = 0–100 ms
>
> For Woody’s algorithm, we expanded the ERP CORE measurement windows by 200 milliseconds in each direction to provide a broader alignment range. The windows used were:
>
> - **N170**: -110–350 ms
> - **MMN**: -125–425 ms
> - **N2pc**: 0–475 ms
> - **N400**: 100–700 ms
> - **P3**: 100–800 ms
> - **LRP**: -300–200 ms
> - **ERN**: -200–300 ms
>
> All baselines were run with the same preprocessing, splits, and evaluation as the main experiments.
>
> | Method                      | n  | K=5               | 10%               | 100%              |
> |-----------------------------|----|-------------------|-------------------|-------------------|
> | Woody's Algorithm           | –  | -424.1 ± 153.9    | -236.3 ± 77.7     | -10.9 ± 15.3      |
> | RIDE                        | –  | -351.5 ± 106.6    | -207.4 ± 54.6     | -29.8 ± 15.9      |
> | Nearest Neighbor            | –  | 14.5 ± 7.0        | 12.0 ± 7.0        | 42.5 ± 4.4        |
> | EEG2ERP                     | 10 | 31.7 ± 0.6        | 36.1 ± 0.5        | 47.7 ± 0.4        |
>
> The results show that Woody and RIDE capture latency jitter but do not effectively denoise the signal. Their estimated ERPs remain noisy and yield negative R squared with only limited smoothing from the low pass filter in RIDE. Nearest Neighbor achieves small but consistent positive R squared that increases with more data, yet remains below EEG2ERP.
>
> To ensure the validity of the implementation we performed tests on a mock dataset of signals with high latency jitter.
>
> The results of RIDE and Woody on the mock dataset are provided here:
> https://i.imgur.com/qIlyMQu.jpeg

---

### Review · Reviewer_Tsca · 2025-08-29

**Summary Of Contributions:**

The paper introduces EEG2ERP, a deep learning approach for estimating ERPs from a smaller number of EEG trials. The strength EEG2ERP method is its performance when only a few EEG trials are available.

**Audience:**

Yes

**Claims And Evidence:**

Yes

**Requested Changes:**

- Ablate the concatenation of the pointwise strandard deviation to the input.
- Please add additional latency-jitter baselines, like RIDE. https://pubmed.ncbi.nlm.nih.gov/25455337/

**Strengths And Weaknesses:**

Strengths:
- Clear writing
- The modeling decisions all make sense
- Experiments are strong for fewer trials
- Methodological contribution is good

Weaknesses:
- I'm not convinced that the "split all trials into two non-overlapping sets for input/output" setting is the best one one. Why not just use the (boostrap) of the non-split full trial as the target? You cited "leakage", but is that the right way to think about this here? If I have a mean of 10 numbers, and I want to use it to approximate the mean of 100 numbers, which include the 10 numbers as a subset, what's the problem?
- Your input is the pointwise-average signal of K trials and K, the number of trials. Then you estimate the variance on the output. I don't understand how it's hypothetically possible to get a good variance estimate without also using the pointwise-variance signal of K trials as well. Is the argument that K is sufficient somehow? Can you please do this experiment and ablate that adding the pointwise-variance (probably point-wise standard deviation in practice for numerical stability) to the input doesn't help?

---

> ### Author Response · Authors · 2025-09-12
>
> We thank the reviewer for their constructive feedback and for highlighting both the strengths of our work and the points where clarification and additional experiments would improve the paper.
>
> ## RE: Trial Split
> Our goal is to evaluate whether a model can predict an event related potential that generalizes to trials that were not used to compute the target. Using the full set average as the target while the model sees a subset introduces dependence between predictor and target. The same single trial fluctuations then appear on both sides of the evaluation. This shared noise inflates correlation and reduces error by construction, which produces an optimistic estimate of performance that does not reflect generalization. Split-half evaluation avoids this optimism as the input half and the target half are disjoint, so any agreement can not arise from shared single trial noise.
>
> A further reason for the use of split-halves is parity of signal to noise ratio. Comparing one half to the other half equalizes the expected variance on both sides and the power of the average. In addition, when the same trials contribute both to the predictor and to the target average, any latency variability in those trials is shared. This overlap dampens the apparent impact of latency variability, because the predictor and the target are influenced by the same misaligned trials. In contrast, in the split-half setting, latency variability is independent across halves. The agreement in this case cannot be explained by shared jitter, so the evaluation is more sensitive to whether a method truly addresses latency variability rather than benefiting from overlap.
>
> Finally, we note that the paper cited by the reviewer (Ouyang et al., 2015, RIDE) also evaluates methods using split-half comparisons.
>
> ## RE: Additional Baselines and Ablations
> We thank the reviewer for the suggestions. As for the point-wise standard deviation, we do not argue that $K$ alone is sufficient. We condition on $K$, but importantly we learn the per–time-point variance via a likelihood loss on *bootstrapped* target ERPs. We argue that this bootstrap-method allows the model to learn good variance estimates. We acknowledge that adding point-wise standard deviation to the input can help when $K$ is large (stable estimates), but in the few-trial regime ($K\sim1$-$5$, our focus) the sample variance is unstable and undefined at $K=1$. We will clarify this in the main paper.
>
> We will be setting up and running additional baselines including the RIDE method and Woody's algorithm. We will provide results on these in the coming days. For the RIDE baseline, we will configure a three-component model representing stimulus, central, and response-related activity ('S', 'C', 'R') and estimate the variable latencies for each component on a single channel basis (the prominent channel).

---

> > ### Comment · Reviewer_Tsca · 2025-09-12
> >
> > Thank you for your response.

---

> ### Author Response · Authors · 2025-09-24
>
> ## RE: Additional baselines
> We implemented Woody's algorithm (Woody 1967), RIDE (Ouyang et al., 2015), and Nearest Neighbor baselines. For RIDE, we used a three component model with stimulus locked (S), central (C), and response locked (R) components, except for LRP and ERN which were modeled with only an R component. The windows were chosen following the RIDE guide for base windows, with the main component window set according to the significant windows reported in ERP CORE. Specifically:
>
> - **N170**: S = 110–150 ms, C = 100–900 ms, R = -300–300 ms
> - **MMN**: S = 125–225 ms, C = 100–900 ms, R = -300–300 ms
> - **N2pc**: S = 200–275 ms, C = 100–900 ms, R = -300–300 ms
> - **N400**: S = 0–500 ms, C = 300–500 ms, R = -300–300 ms
> - **P3**: S = 0–500 ms, C = 300–600 ms, R = -300–300 ms
> - **LRP**: R = -100–0 ms
> - **ERN**: R = 0–100 ms
>
> For Woody’s algorithm, we expanded the ERP CORE measurement windows by 200 milliseconds in each direction to provide a broader alignment range. The windows used were:
>
> - **N170**: -110–350 ms
> - **MMN**: -125–425 ms
> - **N2pc**: 0–475 ms
> - **N400**: 100–700 ms
> - **P3**: 100–800 ms
> - **LRP**: -300–200 ms
> - **ERN**: -200–300 ms
>
> All baselines were run with the same preprocessing, splits, and evaluation as the main experiments.
>
> | Method                      | n  | K=5               | 10%               | 100%              |
> |-----------------------------|----|-------------------|-------------------|-------------------|
> | Woody's Algorithm           | –  | -424.1 ± 153.9    | -236.3 ± 77.7     | -10.9 ± 15.3      |
> | RIDE                        | –  | -351.5 ± 106.6    | -207.4 ± 54.6     | -29.8 ± 15.9      |
> | Nearest Neighbor            | –  | 14.5 ± 7.0        | 12.0 ± 7.0        | 42.5 ± 4.4        |
> | EEG2ERP                     | 10 | 31.7 ± 0.6        | 36.1 ± 0.5        | 47.7 ± 0.4        |
>
> The results show that Woody and RIDE capture latency jitter but do not effectively denoise the signal. Their estimated ERPs remain noisy and yield negative R squared with only limited smoothing from the low pass filter in RIDE. Nearest Neighbor achieves small but consistent positive R squared that increases with more data, yet remains below EEG2ERP.
>
> To ensure the validity of the implementation we performed tests on a mock dataset of signals with high latency jitter.
>
> The results of RIDE and Woody on the mock dataset are provided here:
> https://i.imgur.com/qIlyMQu.jpeg

---

### Decision · Action_Editor_2Lym · 2025-10-16

**Recommendation:** Accept as is

**Additional Comments:**

NA

**Audience:**

Yes

**Audience Explanation:**

The targeted audience is obviously people working on EEG data, and more generally, neuroscientists working on any kind of electrophysiological recordings. The approach will interest people doing inference on signals on multiple, low-SNR and multiple repetitions.

**Claims And Evidence:**

Yes

**Claims Explanation:**

The paper introduces a methodology for predicting event-related potentials (ERPs) of unseen trials using data from just a few initial trials. To accomplish this, it adapts an auto-encoding network that was trained to separate subject- and task-specific information in order to handle ERP inputs and outputs. In addition to estimating the mean, the variance is estimated, and negative log likelihood is used instead of mean squared error for reconstruction. The results show that, with just one or five trials, the model often has a positive R² for predicting the average ERP, which is much better than the baselines.

The authors have strengthened their conclusions with the use of more baseline, the consideration of time structure in the data.
The choice of metrics has been clarified and is overall well-justified now.
The proposed method is not exactly a liver bullet, but the authors do a good job at showing in which situations it is useful.
While the stated goal of the paper may sound a bit narrow, the methodology used ban be at least inspiring for other applications.
Overall, the paper clearly meets expectations for publication in TMLR.